# An arbitrary-spectrum spatial visual stimulator for vision research

Katrin Franke[1,2†], André Maia Chagas[1,3,4†], Zhijian Zhao[1,3],
Maxime JY Zimmermann[4], Philipp Bartel[4], Yongrong Qiu[1,3], Klaudia P Szatko[1,2],
Tom Baden[1,4], Thomas Euler[1,2,3]*

[1]Institute for Ophthalmic Research, University of Tübingen, Tübingen, Germany;
[2]Bernstein Center for Computational Neuroscience, University of Tübingen,
Tübingen, Germany; [3]Center for Integrative Neuroscience, University of Tübingen,
Tübingen, Germany; [4]Sussex Neuroscience, School of Life Sciences, University of
Sussex, Falmer, United Kingdom

**Abstract** Visual neuroscientists require accurate control of visual stimulation. However, few
stimulator solutions simultaneously offer high spatio-temporal resolution and free control over the
spectra of the light sources, because they rely on off-the-shelf technology developed for human
trichromatic vision. Importantly, consumer displays fail to drive UV-shifted short wavelength-
sensitive photoreceptors, which strongly contribute to visual behaviour in many animals, including
mice, zebrafish and fruit flies. Moreover, many non-mammalian species feature more than three
spectral photoreceptor types. Here, we present a flexible, spatial visual stimulator with up to six
arbitrary spectrum chromatic channels. It combines a standard digital light processing engine with
open source hard- and software that can be easily adapted to the experimentalist's needs. We
demonstrate the capability of this general visual stimulator experimentally in the in vitro mouse
retinal whole-mount and the in vivo zebrafish. With this work, we intend to start a community effort
of sharing and developing a common stimulator design for vision research.
DOI: https://doi.org/10.7554/eLife.48779.001

*For correspondence:
thomas.euler@cin.uni-tuebingen.
de

†These authors contributed
equally to this work

Competing interests: The
authors declare that no
competing interests exist.

Reviewing editor: Alexander
Borst, Max Planck Institute of
Neurobiology, Germany

## Introduction

### Challenges in visual stimulation

From psychophysics to single-cell physiology, neuroscientists fundamentally rely on accurate stimulus
control. At first glance, generating visual stimuli appears to be much easier than, for example, olfac-
tory stimuli, because computer screens and video projectors are omnipresent, suggesting a range of
cost-effective choices for the vision researcher. However, commercially available display devices tar-
get human consumers and, thus, are designed for the primate visual system. These devices provide
superb spatial resolution, approximately cover the colour space relevant for trichromatic human
vision (reviewed in *Surridge et al., 2003*) and support refresh rates that consider the human flicker-
fusion frequency (e.g. *Hecht and Verrijp, 1933*). Moreover, as the emphasis is on improving the
subjective viewing experience, commercial display devices typically lack or even purposefully distort
properties that are important when used as visual stimulator for research.

While the spatial resolution provided by even basic commercial displays is typically in excess of
what most model species can resolve, limitations may exist with respect to timing (i.e. refresh rate)
and, in particular, colour space. For example, many insects including *Drosophila* have flicker fusion
frequencies higher than 100 Hz (*Miall, 1978*) and use five or more main visual opsins (*Feuda et al.,
2016*). For most vertebrate model species (e.g. mice and zebrafish), standard refresh rates of ~60 Hz
suffice for the majority of stimulus requirements; however, the limited colour space poses a serious

issue: The light sources (i.e. light-emitting diodes, LEDs) are selected based on the spectrum spanned by the human cone photoreceptor opsins (*Dartnall et al., 1983*; *Nathans et al., 1986*) and spectrally arranged to cover the human trichromatic colour space. Hence, these devices fail to generate adequate colours for species with different spectral photoreceptor sensitivities and typically three-channel devices impose further limitations for species with more than three spectral types of (cone) photoreceptor (reviewed in *Baden and Osorio, 2019*).

Since some of the aforementioned constraints are 'hard-wired' in display devices for the consumer market, it is often impractical if not impossible to modify such devices. Specialised solutions aimed to overcome some of the above constraints are commercially available, for instance, as special calibrated LCD monitors for human psychophysics (e.g. Display++, Cambridge Research Systems, Rochester, UK). However, these solutions are expensive, optimised for primates and often closed source, which makes it difficult for the user to modify them. As a result, vision researchers either invest large amounts of time and/or money aiming to overcome these constraints or are forced to settle on a custom suboptimal solution that addresses the needs of a particular experimental situation. This, in turn, may critically limit the stimulus space that can be routinely explored, and yields substantial problems in reproducibility and interpretation when comparing physiological data between laboratories. Comparability and reproducibility are of particular interest in the backdrop of recent developments in increasingly efficient data acquisition technologies. For example, being able to simultaneously record from hundreds of neurons using multielectrode arrays (e.g. *Jun et al., 2017*) or two-photon functional imaging (e.g. *Ahrens et al., 2013*; *Stringer et al., 2019*) means that experimental limitations are rapidly shifting the 'bottleneck' away from the recording side towards the visual stimulation side.

## Visual stimuli for current animal models

Choosing the adequate animal model for a specific research question may, on the one hand, greatly facilitate the experimental design and the interpretation of the results. On the other hand, when trying to transfer such results to other species, it is critical to keep in mind that each species is adapted to different environments and employs different strategies to survive and procreate (reviewed in *Baden and Osorio, 2019*). In vision research, classical studies often used monkeys and cats as model organisms, which with respect to visual stimuli, for example, in terms of spatial resolution and spectral sensitivity range, have similar requirements as humans. Today, frequently used animal models – such as *Drosophila*, zebrafish or rodents – feature adaptations of their visual systems outside the specifications for human vision: For instance, all of the aforementioned species possess UV-sensitive photoreceptors, zebrafish have tetrachromatic vision, and both zebrafish and *Drosophila* display higher flicker fusion frequencies than most mammals (reviewed in *Marshall and Arikawa, 2014*; *Boström et al., 2016*). Still, many studies in these species use visual stimulation devices produced and optimised for humans. At best, this will suboptimally drive the animal model's visual system, potentially resulting in wrong interpretations of the data.

Here, we present a highly flexible spatial visual stimulator with up to six arbitrary-spectrum chromatic channels. It builds upon a DLP LightCrafter (LCr), which uses a DMD (digital micromirror device) chip, originally developed by Texas Instruments (Dallas, TX; *Hornbeck, 1996*). The LCr and similar DMD-based 'light engines' are broadly used in consumer products. A DMD chip holds an array of tiny mirrors – each representing a pixel – that can be rapidly flipped between two positions, with the 'on' and 'off' position reflecting the incident light towards the projection optics or a 'light dump', respectively. The LCr we used here contains a single DMD chip and, hence, generates colour images by sequentially presenting the R/G/B contents (as $3 \times 8$ bitplanes) of each frame while turning on the respective LED. The intensity of a, say, green pixel is defined by the temporal pattern the corresponding DMD-mirror is flicked between the 'on' and 'off' position while the green LED is constantly on.

Because the spectrum of the light that illuminates the DMD chip can be (almost) arbitrary, a DLP-based projector like the LCr can be customised and adapted to different animal models and their spectral requirements – as has been demonstrated in earlier studies on mice (e.g. *Baden et al., 2016*; *Denman et al., 2017*), zebrafish (e.g. *Guggiana-Nilo and Engert, 2016*) and fruit flies (e.g. *Haberkern et al., 2019*). Alternatively, one can purchase an LCr with custom LEDs (including UV, for commercial systems, see Table 4) or build an external LED unit that illuminates the DMD chip via a light guide (e.g. *Tan et al., 2015*). Here, we use the highly flexible light guide LCr as a light engine

and demonstrate the adaptability of our solution using two exemplary animal species: mice and zebrafish (larvae). Mice currently represent a frequently used model for the mammalian visual system and serve as an example for UV-sensitive vision, while zebrafish are a representative for a well-studied non-mammalian vertebrate species with tetrachromatic vision. Since species-specific chromatic requirements are often more difficult to meet than, for instance, sufficient spatial resolution, our focus here is on adequate chromatic stimulation.

To achieve adequate chromatic stimulation, the spectral composition of the light sources in the stimulator need to cover the spectral sensitivity of the respective model organism. In the ideal case, there should be (*i*) as many LED peaks as the number of spectrally separable photoreceptor types and (*ii*) these should be distributed across the spectral sensitivity range of the species. In general, the spectral sensitivity of an animal is determined by the palette of light-sensitive proteins expressed in their photoreceptors. Vertebrate photoreceptors are divided into rod photoreceptors (rods) and cone photoreceptors (cones). Rods are usually more light-sensitive than cones and, hence, serve vision at dim illumination levels, whereas cones are active at brighter light levels and support colour vision. Depending on the peak sensitivity and the genetic similarity of their opsins, cones are grouped into short (sws, 'S'), medium (mws, 'M') and long wavelength-sensitive (lws, 'L') types, with the sws cones further subdivided into sws1 (near-ultraviolet to blue range, <430 nm, here 'UV') and sws2 (blue range, >430 nm) (reviewed in *Ebrey and Koutalos, 2001*; *Yokoyama, 2000*). The rod:cone ratio of a species is related to the environmental light levels during their activity periods. For instance, while the central fovea of the macaque monkey retina lacks rods altogether, the rod:cone ratio in its periphery is approx. 30:1 (*Wikler and Rakic, 1990*) and therefore similar to that in mice (*Jeon et al., 1998*). In adult zebrafish, the rod:cone ratio is approx. 2:1 (*Hollbach et al., 2015*).

Many vertebrates feature a single type of rod (for exceptions, see *Baden and Osorio, 2019*) but up to five spectral types of cone, which is why cones are more relevant for chromatically adequate visual stimulation. Old-world primates including humans, for example, possess three spectral types of cones (S, M and L). Hence, these primates feature trichromatic daylight vision (reviewed in *Jacobs, 2008*). In contrast, mice are dichromatic like the majority of mammals; they only have two cone types (S and M; *Figure 1a–c*). Unlike most mammals, however, the spectral sensitivities of the mouse are shifted towards shorter wavelengths, resulting in a UV-sensitive S-opsin (*Jacobs et al., 1991*). While one cone type usually expresses only one opsin type, some mammalian species, such as mice or guinea pigs, show opsin co-expression: In mice, for instance, M-cones co-express S-opsin with increasing levels towards the ventral retina (*Figure 1b*) (*Applebury et al., 2000*; *Baden et al., 2013*; *Röhlich et al., 1994*). As a 'more typical' example for non-mammalian vertebrates, the cone-dominated retina of zebrafish contains four cone types, resulting in tetrachromatic vision (*Figure 1d*): In addition to S- and M-cones, they have also UV- and L-cones (*Chinen et al., 2003*). In adult zebrafish, all cone types are organised in a highly regular array, with alternating rows of UV-/S- and M-/L-cones (*Figure 1e,f*) (*Li et al., 2012*). In zebrafish larvae, however, the cone arrangement shows distinct anisotropic distributions for different cone types matched to image statistics present in natural scenes (*Figure 1g,h*) (*Zimmermann et al., 2018*).

Taken together, the diversity of spectral sensitivities present in common animal models used in visual neuroscience as well as their differences to the human visual system necessitates a species-specific stimulator design. Here, we present a highly flexible, relatively low-cost visual stimulation system that combines digital light processing (DLP) technology with easily customisable mechanics and electronics, as well as intuitive control software written in Python. We provide a detailed description of the stimulator design and discuss its limitations as well as possible modifications and extensions; all relevant documents are available online (for links, see *Table 1*). Finally, we demonstrate the use of our stimulator in two exemplary applications; as a dichromatic version for in vitro two-photon (2P) recordings in whole-mounted mouse retina and as a tetrachromatic version for in vivo 2P imaging in zebrafish larvae.

## Results

### Stimulator design

As the 'light engine' of our stimulator, we use the DLP LightCrafter 4500 (here, referred to as 'LCr') developed by Texas Instruments (Dallas, TX). The LCr is a bare-metal version for developers and

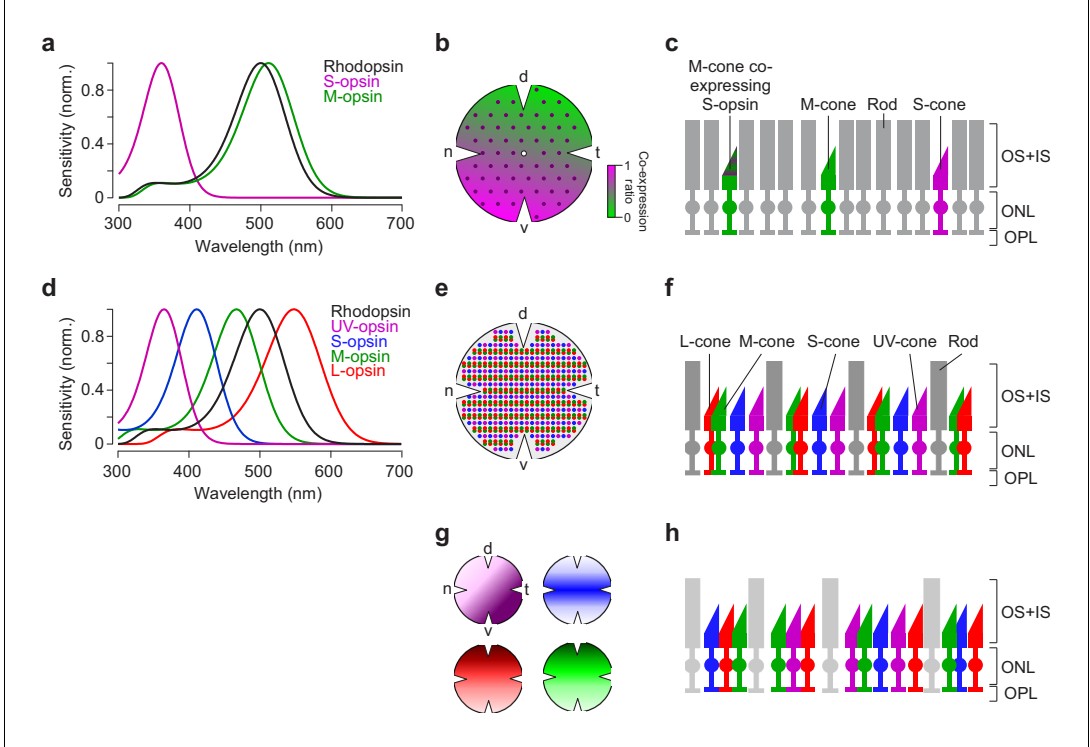

**Figure 1.** Photoreceptor types and distribution in mouse and zebrafish retina. (**a**) Peak-normalised sensitivity profiles of mouse S- (magenta) and M-opsin (green) as well as rhodopsin (black; profiles were estimated following *Stockman and Sharpe, 2000*). (**b**) Schematic drawing of the distribution of cone photoreceptor (cone) types in the mouse; rod photoreceptors (rods) are homogeneously distributed (*Jeon et al., 1998*) (not shown here). Purple dots represent 'true' S-cones exclusively expressing S-opsin (*Haverkamp et al., 2005*); ratio of co-expression of S-opsin in M-cones (*Applebury et al., 2000*; *Baden et al., 2013*) is colour-coded from green to magenta (d, dorsal; t, temporal; v, ventral; n, nasal). (**c**) Illustration of mouse cone and rod arrangement (vertical view; OS+IS, outer and inner segments; ONL, outer nuclear layer; OPL, outer plexiform layer). (**d**) Peak-normalised sensitivity profiles of zebrafish UV- (magenta), S- (blue), M- (green) and L-opsin (red) as well as rhodopsin (black). (**e**) Schematic illustration of the regular cone arrangement in adult zebrafish. Coloured dots represent UV-, S-, M- and L-cones. (**f**) Like (c) but for adult zebrafish retina. (**g**) Schematic drawing illustrating the distribution of cone types in zebrafish larvae (*Zimmermann et al., 2018*). Colours as in (d). (**h**) Like (c,f) for zebrafish larvae. Lighter colour of rods indicate that they are not functional at this age (7–9 dpf; *Branchek and Bremiller, 1984*; *Morris and Fadool, 2005*).
DOI: https://doi.org/10.7554/eLife.48779.002

offers several advantages over consumer devices: (*i*) its control protocol is well documented (for links, see *Table 1*), allowing to program the device via a USB connection on-the-fly; (*ii*) its flexibility in terms of light sources; lightcrafters with customised LEDs (*Table 4*) and a version with a light guide port (see below) are available; (*iii*) its small footprint (15 × 10 × 5 cm) facilitates incorporating the LCr into existing setups. While the stimulators are built around the LCr, we attempted to use a minimum of commercial parts. Except for the specialised optical elements (i.e. dichroic filters, beam splitters, mirrors), most parts can be replaced by 3D printed parts (e.g. designed using OpenSCAD, see Key Resources Table) to increase flexibility and to lower the total costs. For example, instead of commercial rail systems, such as LINOS microbank (Qioptiq, Göttingen, Germany), alternative 3D-printed parts can be used (*Delmans and Haseloff, 2018*). All electronics and the visual stimulation software are Open Source (*Table 1*).

For the two-channel (dichromatic) mouse stimulator (*Figure 2a–c*), we used a light guide LCr (Fiber-E4500MKIITM, EKB Technologies Ltd., Israel). It lacks internal LEDs and the respective beam splitters and instead features a built-in port for a standard light guide (7 mm outer diameter, 5 mm core diameter; see recommendations by EKB). It was coupled by a light guide (for parts list, see *Table 2*) to an external illumination unit (*Figure 2a right,c*). In this unit, a long-pass dichroic mirror combines the light from two band pass-filtered LEDs (with $\lambda_{peak}$ = 387 and 576 nm) and feeds it into a light guide using a fitting collimation adapter. This arrangement facilitates the exchange of the LEDs and allows to mount the illumination unit outside the microscope cabinet. One disadvantage

**Table 1.** For detailed part lists, see *Tables 2* and *3*.

| Part | Links to online resources |
|---|---|
| TI DLP LightCrafter 4500 Evaluation Module ('LCr') | Product overview: http://www.ti.com/tool/dlplcr4500evm User guide: http://www.ti.com/lit/ug/dlpu011f/dlpu011f.pdf Programmer's guide: http://www.ti.com/lit/ug/dlpu010g/dlpu010g.pdf DLP4500 data sheet w/DMD specs: http://www.ti.com/lit/ds/symlink/dlp4500.pdf Alternate DMD windows for increased UV transmission: http://www.ti.com/lit/an/dlpa031d/dlpa031d.pdf DMD reflectance w/o window: https://ntrs.nasa.gov/archive/nasa/casi.ntrs.nasa.gov/20160010355.pdf |
| QDSpy – Open Source Visual stimulation software | Documentation: http://qdspy.eulerlab.de/ Python source code: https://github.com/eulerlab/QDSpy (*Euler, 2019b*; copy archived at https://github.com/elifesciences-publications/QDSpy) Information on 'pattern mode' with QDSpy: http://qdspy.eulerlab.de/lightcrafter.html#example-scripts |
| Chopper | For a 'mechanical' LED blanking solution (see Discussion), based on Thorlabs' Optical Chopper System |
| open-visual-stimulator – Project GitHub repository | Contains spectral calibration scripts, 3D design files for printed parts, printed circuit board design files, bill of materials to populate boards, etc. https://github.com/eulerlab/open-visual-stimulator (*Euler et al., 2019a*; copy archived at https://github.com/elifesciences-publications/open-visual-stimulator) |

DOI: https://doi.org/10.7554/eLife.48779.003

with this current LCr model is, however, that – in our experience – it passes only a fraction of the light entering the light guide port (see Discussion). The LCr is positioned next to the microscope's stage and projects the stimulus via a condenser from below into the recording chamber, where it is focussed on the photoreceptor layer of the isolated mouse retina (*Figure 2a left*). In the type of 2P microscope used here (MOM, Sutter Instruments, Novato, CA; Materials and methods), the scan head including the objective lens – as well as the substage assembly with the condenser – moves relative to the static recording chamber. Hence, to allow the stimulus to 'follow' the objective lens-condenser axis, the LCr is mounted on a pair of low-friction linear slides, with the LCr mechanically coupled to the substage assembly (*Figure 2b*). To allow for stimulus centring, a combination of an x-y and a z-stage, both manually adjustable with micrometer screws, is fitted between slides and LCr.

In addition to this 'through-the-condenser' (TTC) configuration for visual stimulation, we also used the 'through-the-objective' (TTO) configuration described earlier (*Figure 2—figure supplement 1*) (*Euler et al., 2009*). Here, the stimulus is optically coupled into the laser pathway and therefore does not require mechanical coupling of microscope head and visual stimulator. In addition, only scattered light of the visual stimulus will reach the photodetectors above the objective, reducing artefacts caused by stimulus light entering the photodetectors (e.g. photomultipliers, PMTs). However, the disadvantage of the TTO configuration is that the stimulation area is limited by the field-of-view of the objective (approximately 700 µm in diameter for our 20x objective) and, therefore, large-scale retinal networks that may be critical for naturalistic stimulation are likely not well activated.

For the four-channel (tetrachromatic) zebrafish stimulator variant, we optically coupled two light guide LCrs (*Figure 2d,e*; for parts list, see *Table 3*). They used a similar external illumination unit as the mouse stimulator, but with different LED/filter combinations ($\lambda_{peak}$ = 586, 480, 420, and 370 nm). The beams of the two LCrs are collimated and combined using a long-pass dichroic mirror and projected onto a flat teflon screen that covers one side of a miniature water-filled aquarium (*Table 1*), in which the zebrafish larva is mounted on a microscope slide under the objective lens of a MOM-type 2P microscope (*Figure 2e*). Each LCr is placed on an independent three-axis manipulator to facilitate alignment of the two images. Then, small 0.5-cm circular stimuli are projected (one from each stimulator) and the LCrs positions are adjusted using the manipulators until the stimuli are completely overlapping. The general design of the zebrafish stimulator – with one or two LCr

**Table 2.** Parts list of the mouse visual stimulator (*cf.* **Figure 2a–c** and **Figure 2—figure supplement 6**).

| Part | Description (link) | Company | Item number |
| --- | --- | --- | --- |
| Parts for stimulator (except external illumination unit) | | | |
| LCr | 0.45' 'DLP Fiber couple E4500MKII Development Module FC/PC | EKB Technologies Ltd. | DPM-FE4500MKIIF |
| Condenser | C-C Achromat-Aplanat Condenser N.A. 1.40, oil | Nikon | MBL71400 |
| Objective | W Plan-Apochromat 20x/1.0 DIC (UV) VIS-IR | Zeiss | 421452-9880-000 |
| DM 00 | Beamsplitter 900DCXXR | AHF Analysetechnik AG | F73-903 |
| Lens 00 | Achromatic Doublet, f = 75 mm | Thorlabs | AC254-075-A-ML |
| Light guide | Liquid light guide 5 mm Core, 1.2 m length | Thorlabs | LLG05-4H |
| z stage | 13 mm Travel Vertical Translation Stage | Thorlabs | MVS005/M |
| x-y stage | XY Stage, 13 mm Travel | Thorlabs | XYT1/M |
| Frictionless tables | Type NK frictionless table (rollers/balls) | Schneeberger GmbH | NK2-50 (x2) |
| Perfusion chamber | Quick release magnetic imaging chamber | Warner Instruments | 64–1943 |
| Parts for the external illumination unit | | | |
| Cage plate 1 | Cage Plate with Ø2.2' Double Bore | Thorlabs | LCP09/M |
| Cage plate 2 | 30 mm to 60 mm Cage Plate Adapter | Thorlabs | LCP02/M |
| Cage plate 3 | SM1-Threaded Standard Cage Plates | Thorlabs | CP02/M (x7) |
| Cage assembly rods | ER Assembly Rods for 30 mm and 60 mm Cage Systems | Thorlabs | ER1/2/3 (x4) |
| Post holders | Pedestal Post Holders | Thorlabs | PH40E (x2) |
| Post holders | Clamping Forks and Base Adapters | Thorlabs | CF175 (x2) |
| Collimator | 5 mm LLG Collimating Adapter, Zeiss Axioskop | Thorlabs | LLG5A4-A |
| BP 00 | 387/11 BrightLine HC | AHF | F39-387 |
| BP 01 | 576/10 BrightLine HC | AHF | F37-576 |
| UV LED | 385 nm, 320 mW at 350 mA, +- 75° | Roithner | H2A1-H385-r2 |
| Green LED | 590 nm, 8–10 mW at 350 mA, +- 30° | Roithner | M3L1-HY-30 |
| DM 01 | Beamsplitter HC BS 495 | AHF | F38-495 |
| Lens 01 | 1'' N-BK7 Plano-Convex Lens, f = 25.4 mm | Thorlabs | LA1951-A-ML (x2) |
| DM holder | DM_holder, Nylon | Custom-made (University workshop) | |
| LED holder | LED_holder, Aluminium | Custom-made (University workshop) | (x2) |

DOI: https://doi.org/10.7554/eLife.48779.011

projecting onto a teflon screen – is also suitable for visual stimulation during in vivo experiments of other model organisms like mouse or *Drosophila*.

## Separating light stimulation and fluorescence detection

A difficulty when combining visual stimulation with fluorescence imaging is that the spectral photoreceptor sensitivities and the emission spectra of the fluorescent probes tend to greatly overlap. Hence, to avoid imaging artefacts, stimulator light has to be prevented from reaching the PMTs of the microscope, while ensuring that each of the spectral photoreceptor types is stimulated efficiently and as much of the fluorescence signal as possible is captured. To address this issue, light stimulation and fluorescence detection have to be separated temporally and/or spectrally (*Euler et al., 2009*).

**Table 3.** Parts list of the zebrafish visual stimulator (*cf.* **Figure 2d,e**; italic entries are not shown in figure).

| Part | Description (link) | Company | Item number |
|---|---|---|---|
| Parts for stimulator (except external illumination unit) | | | |
| LCr | 0.45' 'DLP Fiber couple E4500MKII Development Module FC/PC | EKB Technologies Ltd. | DPM-FE4500MKIIF (x2) |
| DM 04 | Beamsplitter T 400 LP | AHF | F79-100 |
| Lens 02 | Mounted N-BK7 Bi-Convex Lens, f = 50.0 mm | Thorlabs | LB1844-ML (x2) |
| Lens 03 | Air-Spaced Achromatic Doublet, f = 50 mm | Thorlabs | ACA254-050-A (x3) |
| Lens 04 | Achromatic Doublet, f = 100 mm | Thorlabs | AC508-100-A-ML |
| Light guide | Liquid light guide 5 mm Core, 1.2 m length | Thorlabs | LLG05-4H (x2) |
| *z stage* | *13 mm Travel Vertical Translation Stage* | *Thorlabs* | *MVS005/M (x2)* |
| *x-y stage* | *13 mm Translation Stage* | *Thorlabs* | *MT1B/M (x4)* |
| *Mount plate* | *Aluminium Breadboard* | *Thorlabs* | *MB2530/M* |
| *Lens holder* | *30 mm to 60 mm Cage Plate Adapter* | *Thorlabs* | *LCP02/M (x4)* |
| *Optical Post* | *Optical Post 12.7 mm diam* | *Thorlabs* | *TR100/M (x2)* |
| *Post Holder* | *Post holder 12.7 mm diam 100 mm* | *Thorlabs* | *PH100/M (x2)* |
| *Post clamp* | *Clamping Fork, 1.24' Counterbored Slot* | *Thorlabs* | *CF125-P5 (x2)* |
| *Metal rods* | *Cage Assembly Rod, 12' Long, Ø6 mm* | *Thorlabs* | *ER-12 (x4)* |
| *Metal rods* | *Cage Assembly Rod, 3' Long, Ø6 mm* | *Thorlabs* | *ER-3 (x8)* |
| *Dichroic holder* | *Kinematic Cage Cube Platform for C4W/C6W, Metric* | *Thorlabs* | *B4C/M* |
| *Dichroic holder* | *Cage compatible rectangular filter mount* | *Thorlabs* | *FFM1* |
| *Cage cube* | *60 mm cube cage* | *Thorlabs* | *LC6W* |
| *Lens holder* | *SM1-Threaded 30 mm Cage Plate* | *Thorlabs* | *CP02/M (x2)* |
| *LCr Lens holder* | *LCr lens holder adapter* | *3D printed part* | *(x2)* |
| Fish aquarium | Fish Cinema v10.0_40X_Objective | 3D printed part(s) | |
| Teflon screen | PTFE (Teflon) glass fibre high temperature coating cloth, 0.15 mm | Artistore | |
| Parts for the external illumination units (RGB + UV) | | | |
| Collimator | 5 mm LLG Collimating Adapter, Zeiss Axioskop | Thorlabs | LLG5A4-A (x2) |
| Lens 01 | 1'' N-BK7 Plano-Convex Lens, f = 25.4 mm | Thorlabs | LA1951-A-ML (x4) |
| DM 02 | Laser Beamsplitter H 560 LPXR superflat | AHF | F48-559 |
| DM 03 | Beamsplitter T 450 LPXR | AHF | F48-450 |
| BP 02 | 370/36 BrightLine HC | AHF | F39-370 |
| BP 03 | 420/40 ET Bandpass | AHF | F47-420 |
| BP 04 | 480/40x ET Bandpass | AHF | F49-480 |
| BP 05 | 586/20 BrightLine HC | AHF | F39-587 |
| UV LED | 365 nm 2.4–6.0 mW 20 mA 15° | Roithner | XSL-365-5E |
| Blue LED | 420 nm 420 mW 350 mA 20° | Roithner | SMB1N-420H-02 |

*Table 3 continued*

| Part | Description (link) | Company | Item number |
|------|--------------------|---------|-------------|
| Green LED | 470 nm 70 mW 350 mA 20° | Roithner | SMB1N-D470-02 |
| Red LED | 588 nm 13.5 cd 20 mA 8° | Roithner | B5B-434-TY |
| *Filter/LED/lens holder* | *SM1-Threaded 30 mm Cage Plate* | *Thorlabs* | *CP02/M* (x4) |
| *Collimator holder* | *Double Bore for SM2 Lens Tube Mounting* | *Thorlabs* | *LCP09* (x2) |
| *Vertical holder* | *60 mm to 30 mm Cage System Right-Angle Adapter* | *Thorlabs* | *LCP30* (x3) |
| *Dichroic frame* | *Dichroic frame* | *3D printed part* | (x2) |
| *Frame holder* | *Frame holder* | *3D printed part* | (x2) |
| *Horizontal holder* | *Horizontal holder* | *3D printed part* | (x2) |
| *Metal rods* | *Cage Assembly Rod, 8' Long, Ø6 mm* | *Thorlabs* | *ER-8* (x4) |
| *Metal rods* | *Cage Assembly Rod, 3' Long, Ø6 mm* | *Thorlabs* | *ER-3* (x10) |
| *Optical Post* | *Optical Post 12.7 mm diam* | *Thorlabs* | *TR100/M* (x4) |
| *Post Holder* | *Post holder 12.7 mm diam 100 mm* | *Thorlabs* | *PH100/M* (x4) |
| *Post clamp* | *Clamping Fork, 1.24' Counterbored Slot* | *Thorlabs* | *CF125-P5* (x4) |

DOI: https://doi.org/10.7554/eLife.48779.012

Temporal separation means that the LEDs of the visual stimulator are turned off while collecting the fluorescence signal. In a 'standard' rectangular x-y image scan, the retrace period (when the scanners move to the beginning of the next scan line) can be used for turning the LEDs on to display the stimulus. We found a retrace period of 20% of a scan line (for 1 to 2 ms scan lines) a good compromise between maximising data collection time, avoiding mechanical artefacts from the retracing galvo scanners, and still having sufficient bright stimuli (*Euler et al., 2019*). If other scan patterns are used, LED-on periods (of similar duration as the retrace periods in x-y scans) need to be embedded for the temporal separation concept to work. An example for a more 'mechanical scanner friendly' scan pattern that includes such LED-on periods are the 'spiral' scans we describe elsewhere (*Rogerson et al., 2019*). In any case, the microscope's software has to signal these retrace (or LED-on) periods. Our custom-written microscope software (ScanM, see Materials and methods) generates a 'laser blanking signal' (=low during retrace), which allows turning down the excitation laser's intensity via a Pockels' cell during retrace to reduce the laser exposure of the tissue (*Euler et al., 2019*; *Euler et al., 2009*). Hence, a straightforward way to implement temporal separation between fluorescence detection and light stimulation is to invert this blanking signal and use it to turn on the LEDs during retrace (*Figure 2—figure supplement 2*).

Despite stimulation and data acquisition being temporally separated (see above), spectral separation is needed to protect the PMTs from the stimulus light. Even when the light reaching the PMTs is not bright enough to damage them, it often triggers the overcurrent protection circuit many PMTs are equipped with and shuts them off. Spectral separation is achieved by selecting LED and PMT filters with non-overlapping transmission bands (*Figure 2—figure supplement 3*). This is complemented by a dichroic mirror (DM$_M$ *Figure 2—figure supplements 1* and *3*; DM$_Z$ in *Figure 2—figure supplement 3*) with multiple transmission bands (*Euler et al., 2019*; *Euler et al., 2009*). In the TTO configuration, it transmits one narrow band of stimulation light for each spectral photoreceptor type while reflecting the excitation laser (>800 nm) and the fluorescence signals (detection bands; *Figure 2—figure supplement 3*). In addition, it passes stimulus light reflected back from the specimen. In the TTC configuration, the main role of DM$_M$ is to reflect fluorescence from the specimen to the PMTs while preventing any stimulus light reflected back from the specimen going there by passing it. The same is true for DM$_Z$ in the zebrafish stimulator with the teflon screen, where some of the

stimulus light is scattered in the specimen towards the objective lens. Note that both DMs are not only suitable for mouse and zebrafish but also for other model organisms like *Drosophila* (*Figure 2— figure supplement 4*). An option to further reduce stimulus artefacts that was not evaluated here are gated PMTs (*Euler et al., 2019*).

As explained above, the LCr encodes the brightness of an image pixel by its mirror's 'on' time, and colour sequentially by cycling through the LEDs while presenting the corresponding R-, G- or B-bitplanes in sync (*cf*. LCr User Guide; for link, see *Table 1*). In addition, the LCr allows setting the maximal intensity of each LED via pulse-width modulation (PWM). In the two-channel mouse stimulator, we power the LEDs in the external illumination unit (*Figure 2a* right) using the LCr's onboard LED drivers and therefore, these LEDs are driven as built-in ones would be – except that we interrupt power to the LEDs in sync with the inverted laser blanking signal using a simple custom circuit board (*Figure 2—figure supplement 5a–c*). To switch the necessary currents with sufficient speed, this circuit uses per LED channel three solid state relays connected in parallel. A downside of this simple solution is that the choice of LEDs is constrained by how much current the relays can pass (250 mA continuous current load per relay). The LCr is not limiting here, because its internal LED drivers can provide up to 4.3 A at 5 V in total. For the four-channel zebrafish stimulator, we devised a circuit (logic board, *Figure 2—figure supplement 5d–f*) that uses only the LCr's digital control signals for each LED (LED enable, LED PWM; for details, see *Figure 2—figure supplement 5g*). This board is also compatible with the two-channel mouse stimulator. It represents a more general solution, because it does not rely on power from the LCr. Instead, in combination with custom LED driver boards (*Figure 2—figure supplement 5f*), it can use arbitrary current supplies for the LEDs, making it possible to use any commercially available LED. The logic board supports up to three LED channels, such that for the zebrafish stimulator, two boards are needed (one per LCr) – plus one driver board per LED. For all solutions, printed circuit boards (PCB) designs created using KiCad (see Key Resources Table) and building instructions are provided (see link to repository in *Table 1*).

One potential issue of the described solution for temporal separation is that the frame (refresh) rate of the LCr (typically 60 Hz) and the laser blanking/LED-on signal (500 to 1,000 Hz) are not synchronised and therefore may cause slow aliasing-related fluctuations in stimulus brightness. In practice, however, we detected only small brightness modulations (*Figure 2—figure supplement 2b*).

## Visual stimulation software

Our visual stimulation software (QDSpy) is completely written in Python3 and relies on OpenGL for stimulus rendering. It includes a GUI, which facilitates spatial stimulus alignment, LCr control and stimulus presentation. QDSpy stimuli are written as normal Python scripts that use the 'QDSpy library' to define stimulus objects, set colours, send trigger signals, display scenes etc. Stimulus objects range from simple shapes with basic shader support to videos (for a complete description of the software, see link in *Table 1*). Depending on the way a user implements a stimulus and whether or not the script contains lengthy calculations, it cannot be guaranteed that the script runs fast enough to reliably generate stimulus frames at 60 Hz. Hence, to ensure stimulus timing, the first time QDSpy runs a stimulus script it generates a 'compiled' version of the stimulus, which is stored in a separate file. When the user runs that stimulus again (and the source stimulus script has not been altered after compilation), QDSpy presents the stimulus from the compiled file. This compiled file contains the drawing instructions and timing for every stimulus element used in a very compact form. This strategy has the advantage that stimulus timing is very reliable, as potentially time-consuming sections of the Python stimulus script have already been executed during 'compilation'. The main disadvantage is that user interaction during stimulus presentation is (currently) not possible.

For stimulus presentation, QDSpy relies on the frame sync of the graphics card/driver for stimulus display. By measuring the time required to generate the next frame, the software can detect dropped frames and warn the user of timing inconsistencies, which cannot be altogether excluded on a non-real-time operating system like Windows. Such frame drops, including all other relevant events (e.g. which stimulus was started when, was it aborted etc.) as well as user comments are automatically logged into a file. To account for any gamma correction performed by the LCr firmware when in video mode and/or by non-linearities of the LEDs/LED drivers, we measured each LED's intensity curve separately to generate a lookup table (LUT) that is then used in QDSpy to linearise the colour channels (Materials and methods).

As default, the LCr runs in 'video mode', where it behaves like an HDMI-compatible display (60 Hz, 912 × 1,140 pixels). In this mode, each colour channel in an RGB frame (3 × 8 = 24 bitplanes) is assigned to one of the 3 LCr LEDs via the QDSpy software. It is possible (and supported by QDSpy) to reconfigure the LCr firmware and run it in the so-called 'pattern mode', which, for instance, allows trading bit depth for higher frame rates and assigning each of the 24 bitplanes of every frame to an arbitrary combination of LEDs (Discussion).

The stimulation software generates digital synchronisation markers to align presented stimuli with recorded data. In addition to digital I/O cards (e.g. PCI-DIO24, Measurement Computing, Bietigheim-Bissingen, Germany), QDSpy supports Arduino boards (https://www.arduino.cc/) as digital output device. While the software attempts generating the synchronisation marker at the same time as when presenting the stimulus frame that contains the marker, a temporal offset between these two events in the tens of millisecond range cannot be avoided. We found this offset to be constant for a given stimulation system, but dependent on the specific combination of PC hardware, digital I/O device, and graphic cards. Therefore, the offset must be measured (e.g. by comparing synchronisation marker signal and LCr output measured by a fast photodiode) and considered in the data analysis.

For up to three chromatic channels (e.g. the mouse stimulator, cf. *Figure 2a–c*), stimuli are presented in full-screen mode on the LCr, with the other screen displaying the GUI. When more chromatic channels are needed, as for the zebrafish stimulator, two LCrs are combined (see above; cf. *Figure 2d,e*). QDSpy then opens a large window that covers both LCr 'screens' and provides each LCr with 'its' chromatic version of the stimulus (screen overlay mode). To this end, the software accepts colour definitions with up to six chromatic values and assigns them to the six available LEDs (three per LCr). For example, the first LCr of the zebrafish stimulator provides the red, green and blue channels, whereas the second LCr adds the UV channel (*Figure 2d*). Here, QDSpy presents the stimulus' RGB-components on the half of the overlay window assigned to the first LCr and the stimulus' UV-component on the half of the overlay window assigned to the second LCr. The remaining LED channels are available for a different purpose, such as, for example, separate optogenetic stimulation.

## LED selection and spectral calibration

Adequate chromatic stimulation requires adjusting the stimulator to the spectral sensitivities of the model organism. Ideally, one would choose LEDs that allow maximally separating the different opsins (*Figure 3*). In practice, however, these choices are limited by the substantial overlap of opsin sensitivity spectra (*Figure 3a,c*) and by technical constraints: For instance, commercially available projectors, including the LCr, barely transmit UV light (<385 nm), likely due to UV non-transmissive parts in the optical pathway and/or the reflectance properties of the DMD (Discussion). In addition, when imaging light-evoked neural activity, fluorescence signal detection and visual stimulation often compete for similar spectral bands, and need to be separated to avoid stimulus-related artefacts (*Figure 2—figure supplement 3*; discussed in *Euler et al., 2019*; *Euler et al., 2009*). Compared to projectors with built-in LEDs, the flexible LED complement of the light guide LCr presents a crucial advantage: Here, LEDs can be easily exchanged to avoid the spectral bands of the fluorescent probes, thereby allowing to maximally separate visual stimulation and fluorescence detection.

**Table 4.** Different commercially available UV-enabled projectors.

| Part | Description (link) | Company |
|------|--------------------|---------|
| DPM-E4500UVBGMKII | DLP LightCrafter E4500 MKII with 3 LEDs: UV (385 or 405 nm), blue (460 nm), and green (520 nm) | EKB Technologies Ltd., Bat Yam, Israel https://www.ekbtechnologies.com/ |
| 3DLP9000 UV Light Engine | DLP-based light engine that can be equipped with one arbitrary LED, including UV (365, 385, or 405 nm) | DLi Digital Light innovations, Austin, TX https://www.dlinnovations.com |
| DLP660TE – 4K UHD | 4K-enabled projector, with flexible light sources, using Texas Instruments' DLP660TE chipset | VISITECH Engineering GmbH Wetzlar, Germany https://visitech.no/ |

DOI: https://doi.org/10.7554/eLife.48779.018

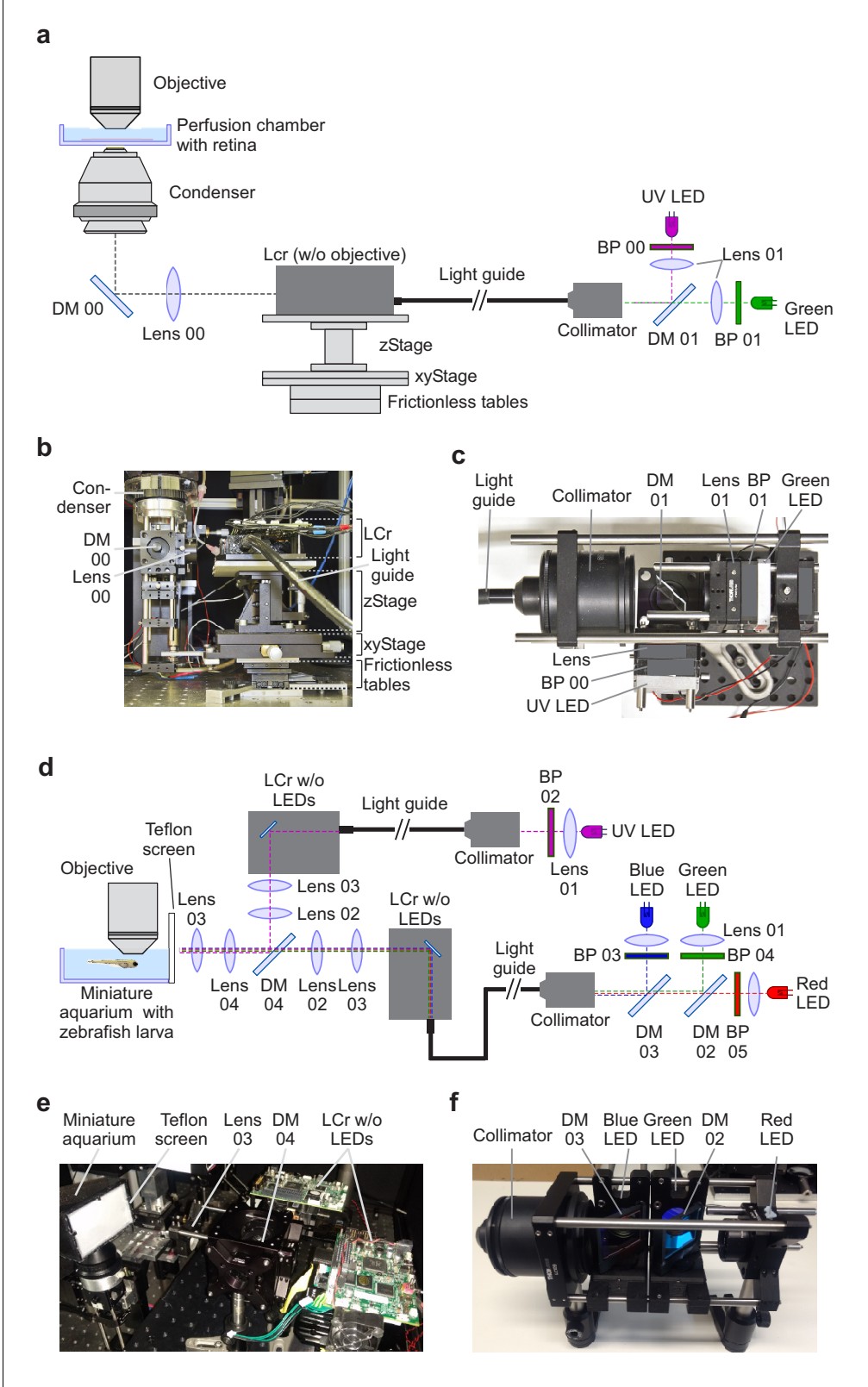

**Figure 2.** Visual stimulator design. (a) Schematic drawing of the dichromatic stimulator for in vitro recordings of mouse retinal explants. The stimulator is coupled into the two-photon (2P) microscope from below the recording chamber with the retinal tissue (through-the-condenser; for alternative light paths (through-the-objective), see *Figure 2—figure supplement 1*). DM, dichroic mirror; BP, band-pass filter; LCr, lightcrafter; LED, light-emitting

*Figure 2 continued on next page*

*Figure 2 continued*

diode. For components, including custom-made parts, see *Table 2*. (b) LCr unit and substage portion of the 2P microscope in side-view. (c) External LED illumination unit in top-view. For details on mechanical parts, see *Figure 2—figure supplement 6*. (d) Schematic drawing of the tetrachromatic stimulator for in vivo recordings in zebrafish larvae. The optical pathways of two LCrs are combined and the stimulus is projected onto a UV-transmissive teflon screen at one side of the miniature aquarium. For components, see *Table 3*. (e) Side-view of tetrachromatic stimulation setup. (f) RGB external LED illumination unit of tetrachromatic stimulation setup. Band-pass (BP) filters 03, 04 and 05 as well as lenses 01 are not indicated due to space constraints.
DOI: https://doi.org/10.7554/eLife.48779.004
The following figure supplements are available for figure 2:

**Figure supplement 1.** Optical pathway for a through-the-objective (TTO) mouse stimulator.
DOI: https://doi.org/10.7554/eLife.48779.005
**Figure supplement 2.** Intensity measurements of the LEDs of the mouse stimulator.
DOI: https://doi.org/10.7554/eLife.48779.006
**Figure supplement 3.** Spectral separation of visual stimulation and fluorescence detection.
DOI: https://doi.org/10.7554/eLife.48779.007
**Figure supplement 4.** Suggestion for LED/filter design for a *Drosophila* visual stimulator.
DOI: https://doi.org/10.7554/eLife.48779.008
**Figure supplement 5.** External LED control and temporal separation of stimulation and fluorescence detection.
DOI: https://doi.org/10.7554/eLife.48779.009
**Figure supplement 6.** Detailed description of external LED unit of the mouse stimulator.
DOI: https://doi.org/10.7554/eLife.48779.010

Because LED spectra can be quite broad, we combine each LED with an appropriate band-pass filter to facilitate arranging stimulation and detection bands.

As a consequence, the peak emissions of the selected LED/filter combinations usually do not match the opsins' sensitivity peaks. For our dichromatic mouse stimulator, we chose LED/filter combinations peaking for UV and green at approximately 385 and 576 nm, respectively (*Figure 3a*), which after calibration (*Figure 3d–f*; Materials and methods), are expected to differentially activate mouse M- and S-opsin (*Figure 3f*). Notably, because of its spectral shift towards shorter wavelengths (*Jacobs et al., 1991*), conventional TFT monitors routinely used in in vivo studies fail to activate mouse S-opsin (*Figure 3b*) and therefore are not able to provide adequate visual stimuli for the mouse visual system (Discussion). For the tetrachromatic zebrafish stimulator, we used LED/filter combinations with peak emissions at approx. 586, 480, 420, and 370 nm (*Figure 3c*). For a suggestion of LED/filter combinations matching the spectral sensitivity of *Drosophila*, see *Figure 2—figure supplement 4*.

To estimate the theoretically achievable chromatic separation of mouse cones with our stimulators, we measured the spectra of each LED/filter combination at different intensities (*Figure 3d*) and converted these data into cone photoisomerisation rates (*Nikonov et al., 2006*). To account for non-linearities in stimulator intensities, we apply gamma correction at the stimulus presentation software level (*Figure 3e*). For our functional recordings (*cf.* Figures 5 and 6), the photoisomerisation rate (in P*/cone/s $\cdot 10^3$) normally ranges from ~0.6 (stimulator shows black image) to ~20 (stimulator shows white image; *Figure 3f*), corresponding to the low photopic regime. In contrast to most commercially available projectors, the current driving the LEDs can also be set to zero, allowing experiments in complete darkness. Further details on the calibration procedures and example calculations for mice and zebrafish are provided in the Methods and in supplemental iPython notebooks, respectively (*Table 1*). Importantly, the general layout of these calibration notebooks facilitates adapting them to other model organisms.

## Spatial resolution

To measure the spatial resolution of our mouse stimulator, we used the 'through-the-objective' (TTO) configuration (*Figure 2—figure supplement 1*) (*Euler et al., 2009*) and projected UV and green checkerboards of varying checker sizes (from 2 to 100 µm; Materials and methods) onto a camera chip positioned at the level of the recording chamber (*Figure 4a*). We found that contrast remained relatively constant for checker sizes down to 4 µm before it rapidly declined (*Figure 4b,c*).

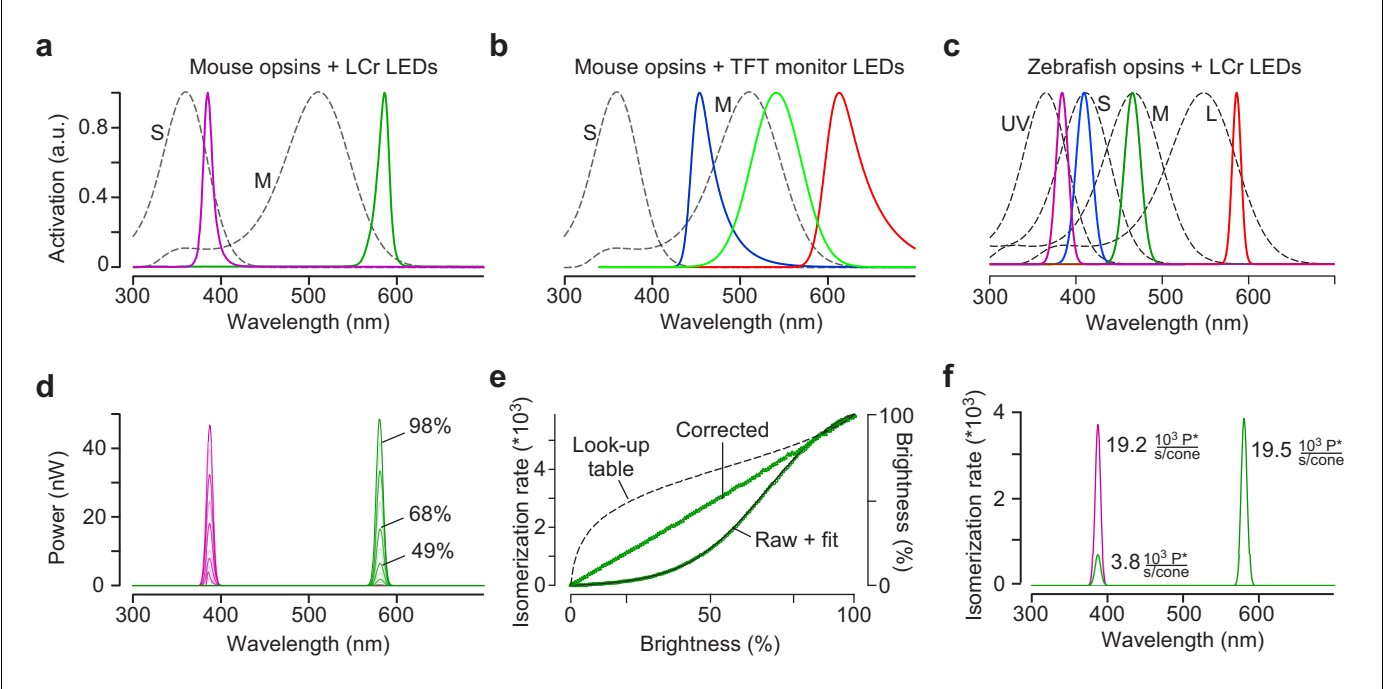

**Figure 3.** Calibration of the mouse stimulator. (**a**) Sensitivity profiles of mouse S- and M-opsin (dotted black lines) and spectra of UV (magenta) and green LED/filter combinations used in the mouse stimulator. (**b**) Sensitivity profiles of mouse S- and M-opsin (dotted black line) and spectra of blue, green and red LED present in a standard TFT monitor. (**c**) Sensitivity profiles of zebrafish opsins (dotted black lines) and spectra of UV, blue, green and red LEDs used in the zebrafish stimulator. (**d**) Spectra (in nW) of UV and green LED obtained from measurements using increasing brightness levels; shown are spectra for 0, 9, 19, 29, 39, 49, 59, 68, 78, 88, and 98% brightness. (**e**) Non-linearised intensity curve ('raw') with sigmoidal fit (black), estimated gamma correction curve (black dotted line; 'Look-up table') and linearized intensity curve ('corrected') for green LED. (**f**) Photoisomerisation rates for maximal brightness of UV (19.2 and 3.8 photoisomerisations $\cdot 10^3$ P*/second/cone for S- and M-opsin, respectively) and green LED (0 and 19.5 $\cdot 10^3$ P*/second/cone for S- and M-opsin, respectively). Note that the UV LED also activates M-opsin due to its increased sensitivity in the short wavelength range (β-band, Discussion).

DOI: https://doi.org/10.7554/eLife.48779.013

Similarly, transitions between bright and dark checkers started to blur for checker sizes below 10 µm (*Figure 4d,e*). For these measurements, we used a 5x objective (MPlan 5X/0.1, Olympus) to project the stimuli, ensuring that the spatial resolution of the camera (OVD5647 chip: 1.4 µm pixel pitch) was not the limiting factor. Hence, a 5 × 5 µm checker stimulus appeared as a 20 × 20 µm square on the camera chip, where it covered approximately $(14.3)^2$ pixels. However, for the scaling factor we use for our recordings (1.9 × 0.9 µm/pixel), a 5 × 5 µm checker consists only of 9.5 × 4.5 LCr pixels (DMD4500, chip area: 6,161 × 9,855 µm with 1,140 × 912 pixels). Thus, the drop in spatial resolution observed for checkers ≤5 µm is likely related to the resolution of the DMD. For the 'through-the-condenser' (TTC) configuration, contrast and sharpness of transitions declined already for checker sizes between 5 and 10 µm (*Figure 4c,e*). That we measured a slightly lower spatial resolution for the TTC compared to the TTO configuration may be because we reached the camera resolution limit (see above), as for TTC we could not simply swap the condenser and, hence, the stimulus image was not magnified on the camera chip.

For the zebrafish stimulator, spatial resolution is less of a problem because, for our setup, checker sizes at the limit of the animal's spatial resolution (2° visual angle; *Haug et al., 2010*) are large (~1 mm on the teflon screen; *cf. Figure 2d,e*).

For the spatial resolution measurements, the UV and green images were each focussed on the camera chip and, therefore, the results do not reflect any effects of chromatic aberration on image quality. To estimate chromatic aberration for our TTO configuration, we next measured the offset between the focal planes of the chromatic channels. Here, we used the standard 20x objective that we also employ for functional recordings. We found that the difference in focal plane between UV and green of approx. 24 µm has little effect on the overall image quality (*Figure 4—figure*

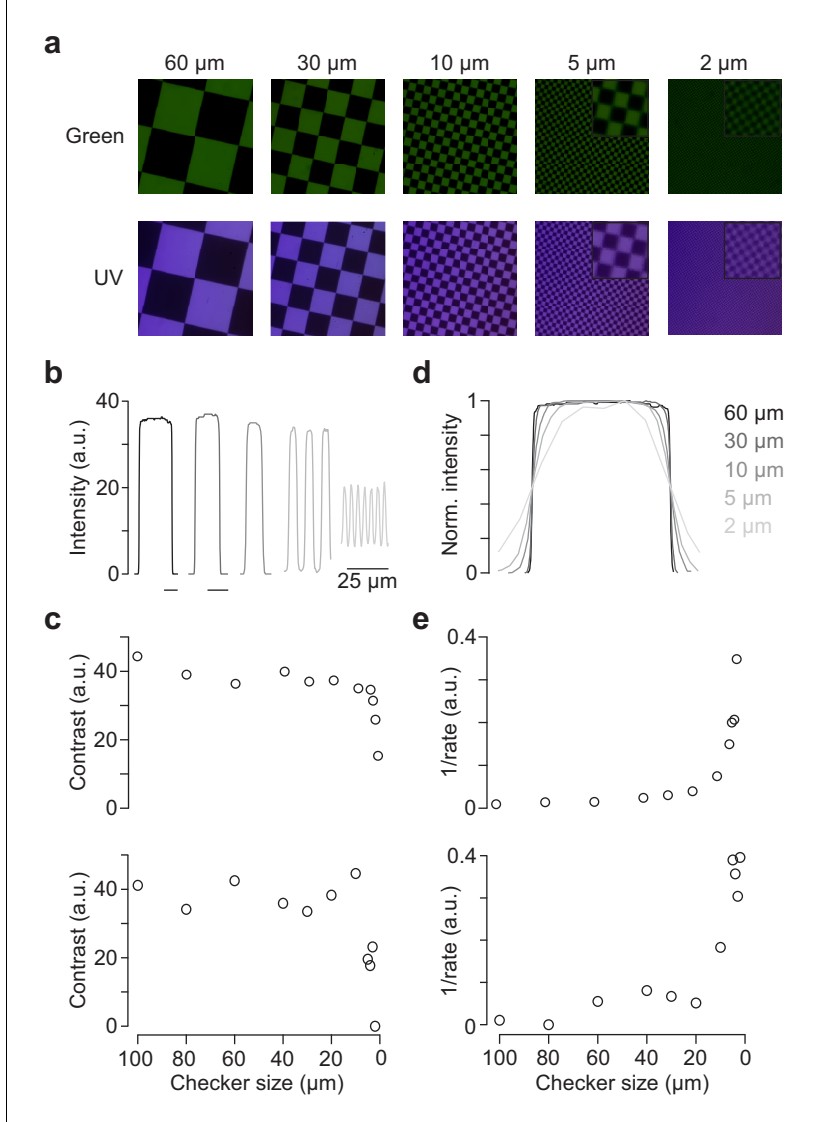

**Figure 4.** Spatial calibration of the mouse stimulator. (**a**) Images of checkerboard stimuli with varying checker sizes projected through-the-objective (TTO) for illumination with green (top) and UV (bottom) LED, recorded by placing the sensor chip of a Raspberry Pi camera at the level of the recording chamber. Focus was adjusted for UV and green LED separately. Insets for 5 and 2 µm show zoomed in regions of the image. (**b**) Intensity profiles for five different checker sizes of green LED. (**c**) Contrasts ($I_{Max} - I_{Min}$) for checkerboards of varying checker sizes of the TTO (top) and through-the-condenser (TTC; bottom) configuration. (**d**) Peak-normalised intensity profiles of different checker sizes, scaled to the same half-maximum width. (**e**) 1/rate estimated from sigmoidal fits of normalised intensity curves like in (d) for varying checker sizes of the TTO (top) and TTC (bottom) configuration.
DOI: https://doi.org/10.7554/eLife.48779.014

The following figure supplement is available for figure 4:

**Figure supplement 1.** Chromatic aberration of the mouse stimulator.
DOI: https://doi.org/10.7554/eLife.48779.015

---

supplement 1) - at least for checker sizes we routinely use for receptive field mapping of retinal neurons (e.g. *Baden et al., 2016*; *cf.* also Discussion).

## Visual stimulation in the explanted mouse retina

To confirm that our stimulator design can be used for adequate chromatic stimulation of the mouse retina, we directly recorded from cone axon terminals in retinal slices using 2P Ca$^{2+}$ imaging (e.g.

*Kemmler et al., 2014*). To this end, we used the transgenic mouse line HR2.1:TN-XL, where the ratiometric $Ca^{2+}$ sensor TN-XL is exclusively expressed in cones (*Figure 5a*) (*Wei et al., 2012*). To quantify the chromatic preference of recorded cones, we calculated spectral contrast (*SC*) based on the response strength to a 1 Hz sine-wave full-field stimulus of green and UV (Materials and methods). The *SC* values correspond to Michelson contrast, ranging from −1 to 1 for the cell responding solely to UV and green, respectively.

In line with the opsin distribution described in mice (*Applebury et al., 2000*; *Baden et al., 2013*), cones located in the ventral retina responded more strongly or even exclusively to UV (*Figure 5b*, bottom row), whereas central cones showed a strong response to both green and UV due to the more balanced co-expression of S- and M-opsin (*Figure 5b*, centre row). In contrast, dorsal cones

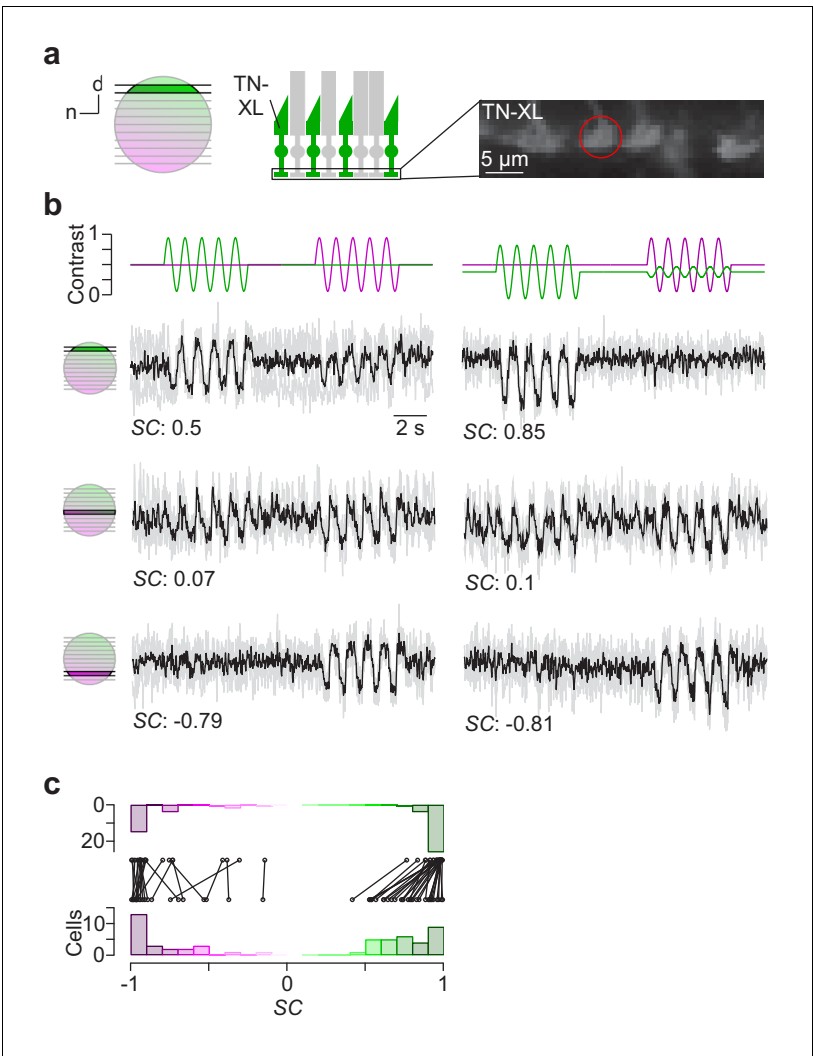

**Figure 5.** Cone-isolating stimulation of mouse cones. (**a**) Dorsal recording field in the outer plexiform layer (OPL; right) shows labelling of cone axon terminals with $Ca^{2+}$ biosensor TN-XL in the HR2.1:TN-XL mouse line (*Wei et al., 2012*). Schematic on the left illustrates retinal location of recorded slice. (**b**) $Ca^{2+}$ traces (mean traces in black, n = 3 trials in grey) of cone axon terminals located in dorsal (top; cone axon terminal from (a)), medial (middle) and ventral (bottom) retina in response to 1 Hz sine modulation of green and UV LED, with spectral contrast (*SC*) indicated below. Colour substitution protocol (right) estimated from calibration data (Materials and methods). (**c**) Distribution and comparison of *SC* for sine modulation stimulus with (top) and without (bottom) silent substitution protocol (n = 55 cells, n = 12 scan fields, n = 1 mouse; p=9.31*10$^{-9}$ for dorsal cells, n = 30; p=0.92 for ventral cells, n = 25; Wilcoxon signed-rank test).
DOI: https://doi.org/10.7554/eLife.48779.016

exhibited a green-dominated response (*Figure 5b*, top row). Due to the cross-activation of M-opsin by the UV LED (see above), most dorsal cones showed an additional small response to UV.

We also tested a stimulus that used silent substitution (*Figure 5b*, right column; Materials and methods) (*Estévez and Spekreijse, 1982*). With this stimulus, we systematically found reduced UV responses in dorsal cones, resulting in a significant shift in *SC* towards more positive values (*Figure 5c*, right column; for statistics, see legend). In contrast, ventral cone responses were not altered by silent substitution.

These data demonstrate that our stimulator design enables obtaining cone-isolating responses in the mouse retina. Notably, the chromatic separation observed in the recordings nicely matches our predictions of cross-activation (see above and Materials and methods).

### Tetrachromatic stimulation in in vivo zebrafish larvae

We recorded in vivo from bipolar cell (BC) axon terminals in zebrafish larvae using 2P Ca$^{2+}$ imaging (*Figure 6a*). The transgenic line we used expressed SyGCaMP6f exclusively in BC axon terminals (*Rosa et al., 2016*). In these experiments, we presented full-field ($90 \times 120$ degrees visual angle) steps or sine wave modulation of red, green, blue and UV light to the teflon screen in front of the immobilised animal (*cf. Figure 2d,e*). This revealed spectrally differential tuning of distinct BC terminals (*Figure 6b,c*), in line with a previous report (*Zimmermann et al., 2018*). For example, terminal one responded with a Ca$^{2+}$ increase to a decrease in red light as well as to an increase in blue or UV light, yielding a 'red$^{Off}$/blue$^{On}$,UV$^{On}$' response behaviour. In contrast, terminal four did not respond to red or green, but differentially responded to blue and UV ('blue$^{Off}$/UV$^{On}$'). Further differences were visible in the temporal profile of the BC responses. For example, terminal three responded more transiently to red and blue, but in a sustained fashion to UV. Similar to cone responses in the in vitro mouse retina, spectrally differential tuning of zebrafish BC terminals was also observed for a sine wave stimulus (*Figure 6c*). Taken together, tetrachromatic stimulation elicited clear differential responses across different wavelengths, thus highlighting that the stimulator's spectral isolation between the four LED channels was sufficient to drive the zebrafish's cone system differentially. To further improve spectral separation, a silent substitution protocol might be used (*cf. Figure 5*; see notebook on GitHub for details). However, as the sensitivity profiles of zebrafish cones substantially overlap (*cf. Figure 1d*), implementing a silent substitution protocol is more challenging than for the mouse.

## Discussion

In this paper, we present a flexible, relatively low-cost stimulator solution for visual neuroscience and demonstrate its use for dichromatic stimulation in the in vitro mouse retina and tetrachromatic stimulation in the in vivo larval zebrafish. The core of the stimulator is an LCr with a light guide port that connects to an external LED array. We also provide detailed calibration protocols (as iPython notebooks) to estimate (cross-)activation in a species' complement of photoreceptor types, which facilitates planning of the LED/filter combinations required for selective chromatic stimulation. To drive the LEDs, we designed simple electronic circuits that make use of the LCr LED control signals and allow integrating an LED-on signal ('blanking signal') for synchronisation with data acquisition, which is critical, for example, for fluorescence imaging in the in vitro retina (*Euler et al., 2009*). By combining two LCrs, up to 6 LED channels are supported by our visual stimulation software (QDSpy). In addition, we describe three exemplary projection methods that allow tuning the system towards high spatial resolution ('through-the-objective') or a large field-of-stimulation ('through-the-condenser') for in vitro experiments, or presentation on a teflon screen for in vivo studies. All materials (electronics, optical design, software, parts lists etc.) are publically available and open source.

### The need for 'correct' spectral stimulation

The spectral sensitivity markedly varies across common model organisms used in visual neuroscience (*cf*. Introduction). As a result, in most cases visual stimulation devices optimised for the human visual system do not allow 'correct' spectral stimulation, in the sense that the different photoreceptor types are not differentially activated by the stimulator LEDs. Instead, 'correct' spectral stimulation requires that the visual stimulator is well-adjusted to the specific spectral sensitivities of the model organism.

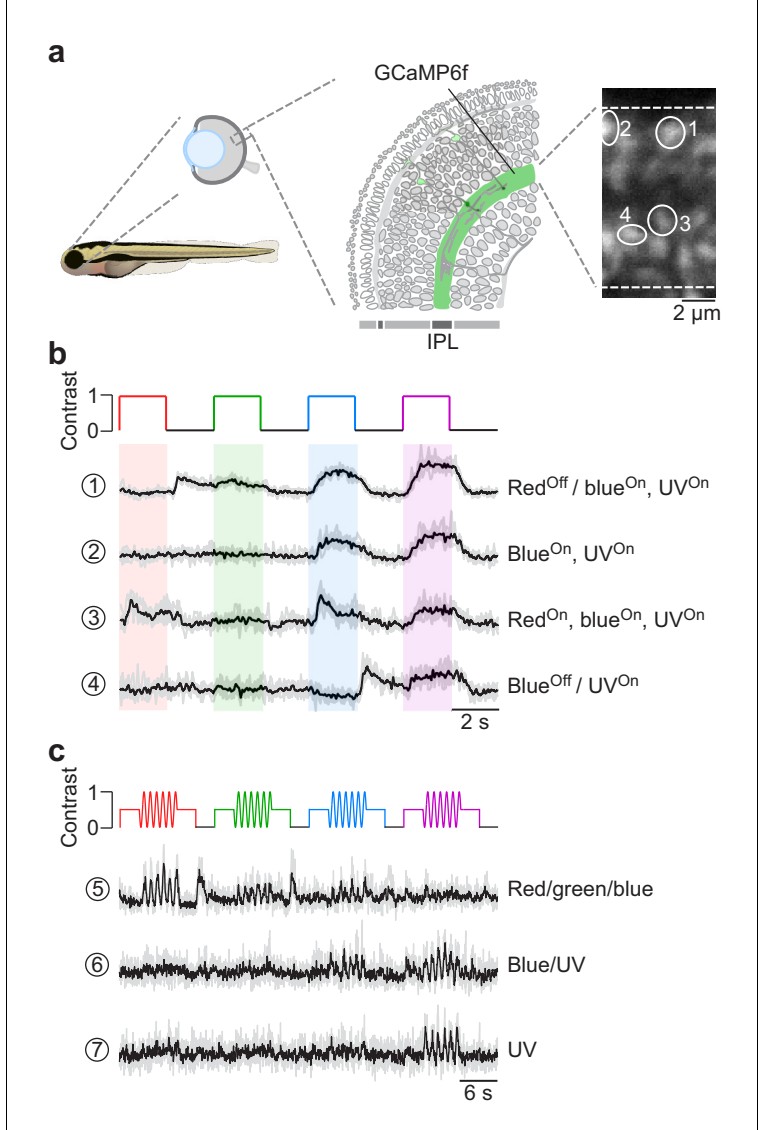

**Figure 6.** Chromatic responses in bipolar cells of in vivo zebrafish larvae. (a) Drawing illustrating the expression of the genetically encoded $Ca^{2+}$ biosensor SyGCaMP6f in bipolar cell terminals (left) of *tg(1.8ctbp2:SyGCaMP6f)* zebrafish larvae and scan field of inner plexiform layer (IPL; right), with exemplary regions-of-interest (ROIs) marked by white circles. (b) Mean $Ca^{2+}$ traces (black; n = 6 trials in grey) in response to red, green, blue and UV full-field flashes (90 × 120 degrees visual angle, presented to the fish's right side). (c) Mean $Ca^{2+}$ traces (black; n = 4 trials in grey) in response to full-field sine modulation (at 1 Hz) of red, green, blue and UV LED.
DOI: https://doi.org/10.7554/eLife.48779.017

For example, while human S-opsin is blue-sensitive (reviewed in *Jacobs, 2008*), the S-opsin of mice shows its highest sensitivity in the UV range (*Figure 1a*) (*Jacobs et al., 1991*). As standard TFT monitors optimised for humans and routinely used in mouse in vivo studies do not emit in the UV range, they fail to activate mouse S-opsin (*cf. Figure 3b*). If then, due to space constraints, the stimulation monitors are positioned in the UV-sensitive upper visual field of the mouse (*cf. Figure 1b*), such a stimulator will mainly activate the rod pathway. As a result, the presentation of 'truly' mouse-relevant natural stimuli is hampered, if not impossible. In recent years, however, several studies used customised projectors that allow UV stimulation for investigating chromatic processing in, for example, dLGN (*Denman et al., 2017*) or V1 (*Tan et al., 2015*). Here, similar to the arrangement in our zebrafish stimulator (*cf. Figure 2d*), the image is either back-projected onto a UV-transmissive teflon

screen (*Tan et al., 2015*) or projected onto a visual dome coated with UV-reflective paint (*Denman et al., 2017*). Both solutions are compatible with the mouse stimulator described above.

Even when the stimulator is adjusted to the spectral sensitivity of the model organism, each stimulator LED typically activates more than one photoreceptor type due to overlapping sensitivity profiles of the different opsins (*cf. Figure 1*). In particular, the long sensitivity tail of opsins for shorter wavelengths ('β-band') contributes to cross-activation of photoreceptors by the stimulator LEDs. For example, the sensitivity of mouse M-opsin to our UV LED results in a cross-activation of ~19.5% (*cf. Figure 3f*). Such 'imperfect' spectral separation of cone types is sufficient to investigate many questions concerning chromatic processing in the visual system – especially as there rarely is photoreceptor type-isolating stimulation in natural scenes (*Chiao et al., 2000*). If needed, photoreceptor cross-activation can be ameliorated by using a silent substitution protocol (*Estévez and Spekreijse, 1982*; but see *Kamar et al., 2019*). Here, one type of photoreceptor is selectively stimulated by presenting a steady excitation to all other photoreceptor types using a counteracting stimulus (*cf. Figure 5b*). This allows, for instance, to investigate the role of individual photoreceptor types in visual processing.

## Stimulation with UV light

Sensitivity to UV light is widespread across animal species (reviewed in *Cronin and Bok, 2016*). Sometimes UV sensitivity may represent a specialised sensory channel; for example many insects and potentially some fish use UV-sensitive photoreceptors to detect polarisation patterns in the sky for orientation (*Parkyn and Hawryshyn, 1993*; *Seliger et al., 1994*; *Wehner, 2001*). In most cases, however, UV sensitivity seems to be simply incorporated into colour vision, extending the spectral range accessible to the species. Here, UV sensitivity can play an important role in invertebrate and vertebrate behaviour, including navigation and orientation, predator and prey detection, as well as communication (reviewed in *Cronin and Bok, 2016*).

For instance, mice possess a UV-sensitive S-opsin, which is co-expressed by M-cones predominantly in the ventral retina (; *Baden et al., 2013*). As the ventral retina observes the sky, it was proposed that the ventral UV sensitivity promotes detection of predatory birds, which appear as dark silhouettes against the sky. As UV light dominates the (clear) sky due to increased Rayleigh scattering of short wavelengths, contrasts tend to be higher in the UV channel (discussed in *Applebury et al., 2000*; *Baden et al., 2013*; *Cronin and Bok, 2016*). In support of this, recordings from mouse cones suggest that ventral S/M-cones prefer dark contrasts, whereas dorsal M-cones encode bright and dark contrasts symmetrically (*Applebury et al., 2000*; *Baden et al., 2013*).

Zebrafish larvae express the UV-sensitive sws2 opsin in their UV-cones. UV-vision in zebrafish is likely used for several tasks, including prey detection, predator and obstacle avoidance as well as colour vision (*Yoshimatsu et al., 2019*; *Zimmermann et al., 2018*). Like in mice, the distribution of UV-cones is non-uniform across the retinal surface. UV-cone density is highest in the temporo-ventral retina which surveys the upper-frontal part of visual space. This UV-specific *area centralis* is likely a specialisation for prey capture: Larval zebrafish feed on small, water-borne microorganisms such as paramecia, which are largely translucent at long wavelengths of light but readily scatter UV (*Johnsen and Widder, 2001*; *Novales Flamarique, 2013*). Next, unlike for most terrestrial animals, predators may appear in any part of visual space in the aquatic environment, and zebrafish invest in UV-dark detection of predator-silhouettes throughout visual space (*Losey et al., 1999*; *Yoshimatsu et al., 2019*; *Zimmermann et al., 2018*). Finally, UV-sensitivity is integrated into retinal circuits for colour vision to impact tetrachromatic vision, as originally demonstrated for goldfish (*Neumeyer, 1992*).

Taken together, to approach natural conditions when probing a UV-sensitive species' visual system, UV stimulation must be included. Nonetheless, there are some pitfalls specifically linked to UV light stimulation. One major issue is that, in our experience, the standard LCr barely transmits wavelengths <385 nm. As the reflectance of the micromirrors (aluminium) drops only <300 nm and the glass window covering the DMD transmits ≥90% of the light down to 350 nm (see links in *Table 1*), one limiting factor appears to arise from the LCr optics. Therefore, if shorter wavelengths are required, replacing the internal optics of the projector is necessary (e.g. *Tan et al., 2015*). If the different stimulation wavelengths are spread across a large range (e.g. $\Delta\lambda$ = 191 and 200 nm for zebrafish and mouse stimulator, respectively; *cf. Figure 3a,c*), chromatic aberration may become an issue, causing an offset between the focal planes of the different colour channels (*cf. Figure 4—figure*

*supplement 1*). For our TTO stimulator configuration, we found a focus difference between UV and green in the order of a few tens of micrometers. For a checker size that is commonly used for receptive field mapping of retinal neurons (e.g. 40 µm; *Baden et al., 2016*), we observed only a slight image blurring due to chromatic aberration, that likely has a negligible effect on our experiments. If chromatic aberration becomes an issue, viable approaches may be to increase the depth-of-field (e.g. by decreasing the aperture size with a diaphragm in the stimulation pathway) and/or use appropriate achromatic lenses.

## Potential issues and technical improvements

In this section, we discuss potential issues that may arise when adapting our stimulator design to other experimental situations, as well as possible technical improvements.

If too much stimulation light enters the PMTs, in addition to spectral separation also temporal separation of visual stimulation and data acquisition is needed. To address this problem, we presented here an electronic solution that allows the LEDs to be on only during the short retrace period of a scan line. However, if higher LED power and/or shorter LED-on intervals are needed, the design of the 'blanking' circuits becomes more challenging, because handling fast switching of high currents and voltages with short rise and decay times is difficult (e.g. see our 'driver' boards on GitHub). Here, an alternative is to use a mechanical chopper (see *Table 1* and *Alfonso-Garcia et al., 2019*; *Yang et al., 2019*). Briefly, a custom 3D printed chopping blade is attached to the chopper and the system is mounted at an appropriate position in the light path such that the blade is able to block the stimulus during the system's scanning period. The blanking signal from the microscope software (see Results) is used to synchronise chopper rotation speed. The main advantage of this solution is that it works with any stimulator and without meddling with its electronics. Disadvantages include, however, (i) mechanical vibrations and spinning noise, (ii) that different scanning modes require different chopping blades, and (iii) the additional costs for the chopper.

For increased flexibility with respect to the LED complement of the visual stimulator, we here use an external LED unit coupled into the LCr via a light guide port (*cf. Figure 2*). One disadvantage with this LCr model is, however, that it passes only a relatively small fraction of the light entering the light guide port. While this is not problematic for small projection areas used in our mouse and zebrafish recordings or for relatively low light intensities, it may become an issue when projecting the stimulus onto a larger area like the inside of a dome (e.g. *Denman et al., 2017*; *Schmidt-Hieber and Häusser, 2013*). Here, LCr models with built-in, high-power LEDs might be a better option (*Table 4*; *Supplementary file 1*).

If high spatial resolution is not required, an interesting alternative to a projector-based stimulator is one built from arrays of LEDs (e.g. *Reiser and Dickinson, 2008*; *Wang et al., 2019*). The main advantage of LED arrays is that they offer a more precise timing control compared to the combination of HDMI display and PC graphics card driven by software running on a desktop PC. Hence, LED arrays may allow refresh rates in the range of several hundreds of Hz (*Reiser and Dickinson, 2008*). However, apart from their lower spatial resolution, current LED arrays only support a low number of colour channels, making them less well suitable for chromatic processing studies. In addition, LED arrays typically require customised control electronics, whereas stimulators based on standard HDMI displays can be driven by the experimenter's software of choice.

For the experiments shown here, we run the LCr for simplicity in 'video mode', where it acts as a normal 60 Hz HDMI display. It is also possible to configure the LCr's firmware in 'pattern mode' without requiring changes to the stimulator hardware. In pattern mode, the user can precisely define how the incoming stream of RGB bitplanes is interpreted and displayed. For example, it is possible to assign multiple LEDs to individual bitplanes and combinations thereof. Moreover, if a lower bit depth is acceptable, much higher frame rates can be achieved. While QDSpy supports the pattern mode, stimulus design can be more challenging, because the LCr receives its video input as 24 bit RGB data frames at a rate of 60 Hz, no matter how pattern mode interprets these data. For example, when configuring the LCr for 120 Hz (at half the bit depth), two consecutive 'display frames' need to be encoded in one 24-bit data frame by the stimulation software. Hence, the user should have a thorough knowledge of the LCr's design, and studying the documentation of the LCr's programming interface provided by Texas Instruments is recommended (for further details, see links in *Table 1*).

### Towards a common stimulator design for vision research

Visual neuroscientists fundamentally rely on accurate stimulation, which not only includes choosing the appropriate spatial and temporal resolution, but also free control over the spectral properties. However, unlike for other equipment, such as amplifiers for electrophysiology, there is no selection of 'standardised' stimulation devices. Due to the lack of standardised yet flexible visual stimulators, many vision researchers employ specialised, often incompletely described solutions that address the needs of a particular experiment. This, in turn, may yield substantial problems in reproducibility and interpretation when comparing physiological data between laboratories. Here, we provide a detailed description of a highly flexible visual stimulator solution that uses commercial hardware only where necessary and otherwise relies on open hard- and software components. As our design can be easily adapted to different species' and the experimentalist's needs, it is suitable for a wide range of applications, ranging from psychophysics to single-cell physiology. By combining two LCrs and running them in pattern mode, structured stimulation with up to six chromatic channels at high frame rates is possible, which is critical for species with more than three spectral photoreceptor types and higher flicker fusion rates, such as many insects.

With this paper, we intend to start a community effort of sharing and further developing a common stimulator design for vision science. As the programming interface of the used DLP engine is publicly available, our system can also serve as a useful starting point for further community developments. To foster interactions, we set up a public GitHub repository inviting other vision researchers to use, share, and improve our design. With this effort, we hope to increase comparability and reproducibility of data in the field of visual neuroscience across labs.

## Materials and methods

**Key resources table**

| Reagent type (species) or resource | Designation | Source or reference | Identifiers | Additional information |
|---|---|---|---|---|
| Genetic reagent (*Mus musculus*) | HR2.1:TN-XL | *Wei et al., 2012* | | Dr. Bernd Wissinger (Tübingen University) |
| Genetic reagent (*Danio rerio*) | tg(1.8ctbp2:SyGCaMP6f) | *Rosa et al., 2016* | | Dr. Leon Lagnado (Sussex University) |
| Software, algorithm | KiCad EDA | http://kicad-pcb.org/ | | Electronics design software |
| Software, algorithm | OpenSCAD | http://www.openscad.org | | 3D CAD software |

Note that the general stimulator design, operation and performance testing is described in the Results section. The respective parts for the mouse and the zebrafish stimulator versions are listed in *Tables 2* and *3*, respectively. Hence, this section focuses on details about the calibration procedures, 2P imaging, animal procedures, and data analysis.

### Intensity calibration and gamma correction

The purpose of the intensity calibration is to ensure that each LED evokes a similarly maximal photo-isomerisation rate in its respective spectral cone type, whereas the gamma correction aims at linearising each LED's intensity curve. All calibration procedures are described in detail in the iPython notebooks included in the open-visual-stimulator GitHub repository (for link, see *Table 1*).

In case of the **mouse stimulator**, we used a photo-spectrometer (USB2000, 350–1000 nm, Ocean Optics, Ostfildern, Germany) that can be controlled and read-out from the iPython notebooks. It was coupled by an optic fibre and a cosine corrector (FOV 180°, 3.9 mm aperture) to the bottom of the recording chamber of the 2P microscope and positioned approximately in the stimulator's focal plane. For intensity calibration, we displayed a bright spot (1,000 μm in diameter, max. intensity) of green and UV light to obtain spectra of the respective LEDs. We used a long integration time (1 s) and fitted the average of several reads (n = 10 for green; n = 50 reads for UV) with a Gaussian to remove shot noise. This yielded reliable measurements also at low LED intensities, which was particularly critical for UV LEDs.

The spectrometer output ($S_{meas}$) was divided by the integration time ($\Delta t$, in s) to obtain counts/s and then converted into electrical power ($P_{el}$, in nW) using the calibration data ($S_{Cal}$, in μJ/count) provided by Ocean Optics,

$$P_{el}(\lambda) = S_{Meas}(\lambda)/\Delta t \cdot S_{Cal}(\lambda) \cdot 10^3, \tag{1}$$

with wavelength $\lambda$. To obtain the photoisomerisation rate per photoreceptor type, we first converted from electrical power into energy flux ($P_{eflux}$, in eV/s),

$$P_{eflux}(\lambda) = P_{el}(\lambda) \cdot a \cdot 10^{-9}, \tag{2}$$

where $a$ = 6.242 $\cdot 10^{18}$ eV/J. Next, we calculated the photon flux ($P_{Phi}$, in photons/s) using the photon energy Q ($P_Q$, in eV),

$$P_Q(\lambda) = c \cdot h / \left(\lambda \cdot 10^{-9}\right), \tag{3}$$

$$P_{Phi}(\lambda) = P_{eflux}(\lambda) / P_Q(\lambda), \tag{4}$$

with the speed of light, $c$ = 299,792,458 m/s, and Planck's constant, $h$ = 4.135667 $\cdot 10^{-15}$ eV·s. The photon flux density ($P_E$, [photons/s/μm$^2$]) was then computed as

$$P_E(\lambda) = P_{Phi}(\lambda) / A_{Stim}, \tag{5}$$

where $A_{Stim}$ (in μm$^2$) corresponds to the light stimulus area. To convert $P_E$ into photoisomerisation rate, we next determined the effective activation ($S_{Act}$) of mouse photoreceptor types by the LEDs as

$$S_{Act}(\lambda) = S_{Opsin}(\lambda) \cdot S_{LED}(\lambda) \tag{6}$$

with the peak-normalised spectra of the M- and S-opsins, $S_{Opsin}$, and the green and UV LEDs, $S_{LED}$. Sensitivity spectra of mouse opsins were derived from Equation 8 in *Stockman and Sharpe (2000)*.

For our LEDs (*Table 2*), the effective mouse M-opsin activation was 14.9% and 10.5% for the green and UV LED, respectively. The mouse S-opsin is only expected to be activated by the UV LED (52.9%) (*Figure 3a, f*). Next, we estimated the photon flux ($R_{Ph}$, [photons/s]) for each photoreceptor as

$$R_{Ph}(\lambda) = P_E(\lambda) \cdot A_{Collect} \tag{7}$$

where $A_{Collect} = 0.2$ μm$^2$ corresponds to the light collection area of cone outer segments (*Nikonov et al., 2006*). The photoisomerisation rate ($R_{Iso}$, P*/photoreceptor/s) for each combination of LED and photoreceptor type was estimated using

$$R_{Iso} = \sum R_{Ph}(\lambda) \cdot S_{Act}(\lambda), \tag{8}$$

see *Nikonov et al. (2006)* for details. The intensities of the mouse stimulator LEDs were manually adjusted (*Figure 2—figure supplement 5b,c*) to an approximately equal photoisomerisation range from (in P*/cone/s $\cdot 10^3$) 0.6 and 0.7 (stimulator shows black image) to 19.5 and 19.2 (stimulator shows white image) for M- and S-opsins, respectively (*cf. Figure 3f*). This corresponds to the low photopic range. The M-opsin sensitivity spectrum displays a 'tail' in the short wavelength range (due to the opsin's β-band, see *Figure 1a* and *Stockman and Sharpe, 2000*), which means that it should be cross-activated by our UV LED. Specifically, while S-opsin should be solely activated by the UV LED (19.2 by UV vs. 0.1 by green; in P*/cone/s $\cdot 10^3$), we expect M-opsin to be activated by both LEDs (19.5 by green vs. 3.8 by UV). The effect of such cross-activation can be addressed, for instance, by silent substitution (see below).

To account for the non-linearity of the stimulator output using gamma correction, we recorded spectra for each LED for different intensities (1,000 μm spot diameter; pixel values from 0 to 254 in steps of 2) and estimated the photoisomerisation rates, as described above. From these data, we computed a lookup table (LUT) that allows the visual stimulus software (QDSpy) to linearise the intensity functions of each LED (*cf. Figure 3e*; for details, see iPython notebooks; *Table 1*).

In case of the **zebrafish stimulator**, to determine the LED spectra, we used a compact CCD Spectrometer (CCS200/M, Thorlabs, Dachau, Germany) in combination with the Thorlabs Optical Spectrum Analyzers (OSA) software, coupled to a linear fibre patch cable. To determine the electrical power ($P_{el}$, in nW), we used an optical energy power meter (PM100D, Thorlabs) in combination with the Thorlabs Optical Power Monitor (OPM) software, coupled to a photodiode power sensor (S130VC, Thorlabs). Both probes were positioned behind the teflon screen (0.15 mm, for details, see *Table 3*). Following the same procedure as described above, we determined the photoisomerisation rate ($R_{Iso}$, P*/photoreceptor/s) for each combination of LED and photoreceptor type (*cf.* iPython notebooks; *Table 1*).

## Spatial resolution measurements

To measure the spatial resolution of the mouse stimulator, we removed the lens of a Raspberry Pi camera chip (OV5647, Eckstein GmbH, Clausthal-Zellerfeld, Germany) and positioned it at the level of the recording chamber. Then, we projected UV and green checkerboards of varying checker sizes (2, 3, 4, 5, 10, 20, 30, 40, 60, 80, and 100 µm) through an objective lens (MPL5XBD (5x), Olympus, Germany) or through the condenser onto the chip of the camera (*Figure 4a*). For each checker size and LED, we extracted intensity profiles using ImageJ (*Figure 4b*) and estimated the respective contrast as $I_{Max} - I_{Min}$ (*Figure 4c*). To quantify the steepness of the transition between bright and dark checkers, we peak-normalised the intensity profiles and normalised relative to half-width of the maximum (*Figure 4d*). Next, we fitted a sigmoid to the rising phase of the intensity profile

$$y = K_0 + K_1 / (1 + exp(-(x - K_2)/K_3)) \tag{9}$$

and used $1/K_3$ as estimate of the rise time and as a proxy for 'sharpness' of the transitions between black and white pixels (*Figure 4e*).

To measure the difference in focal plane of UV and green LED due to chromatic aberration, we projected a 40 and 100 µm checkerboard through a 20x objective (W Plan-Apochromat 20×/1.0 DIC M27, Zeiss, Oberkochen, Germany) onto the Raspberry Pi camera (see above).

## Fast intensity measurements

To verify temporal separation, we measured the time course of the green LED (mouse stimulator) with and without blanking using a PMT positioned at the level of the recording chamber (*Figure 2— figure supplement 2a*). Traces were recorded with pClamp at 250 kHz (Molecular Devices, Biberach an der Riss, Germany). To estimate the amount of intensity modulation due to aliasing (*Figure 2— figure supplement 2b*), we measured the intensity of both LEDs together ('white') driven by a chirp stimulus with blanking at the same position using a photodiode (Siemens silicon photodiode BPW 21, Reichelt, Sande, Germany; as light-dependent current source in a transimpedance amplifier circuit). Next, intensity traces were box-smoothed with a box width of 100 ms, which roughly corresponds to the integration time of mouse cone photoreceptors (*Umino et al., 2008*).

## Silent substitution

For our measurements in mouse cones, we used a silent substitution protocol (*Estévez and Spekreijse, 1982*) for generating opsin-isolating stimuli to account for the cross-activation of mouse M-opsin by the UV LED. Here, one opsin type is selectively stimulated by presenting a scaled, counterphase version of the stimulus to all other opsin types (*cf. Figure 5*). Specifically, we first used the ratio of activation (as photoisomerisation rate) of M-opsin by UV and green to estimate the amount of cross-activation ($S_{CrossAct}$). For our recordings, an activation of M-opsin of 19.5 and 3.8 P*/cone/s $\cdot 10^3$ for green and UV LED resulted in a cross-activation of $S_{CrossAct} = 0.195$. Then, $S_{CrossAct}$ was used to scale the intensity of the counterphase stimulus:

$$I_G = I - I_{UV} \cdot S_{CrossAct} \tag{10}$$

For our recordings in zebrafish larvae, we did not use silent substitution. However, we describe a possible approach for the zebrafish (or a comparable tetrachromatic species) in our online resources (*Table 1*).

## Animals and tissue preparation

All animal procedures (mice) were approved by the governmental review board (Regierungspräsidium Tübingen, Baden-Württemberg, Konrad-Adenauer-Str. 20, 72072 Tübingen, Germany) and performed according to the laws governing animal experimentation issued by the German Government. All animal procedures (zebrafish) were performed in accordance with the UK Animals (Scientific Procedures) Act 1986 and approved by the animal welfare committee of the University of Sussex (zebrafish larvae).

For the **mouse** experiments, we used one 12-week-old HR2.1:TN-XL mouse; this mouse line expresses the ratiometric $Ca^{2+}$ biosensor TN-XL under the cone-specific HR2.1 promoter and allows measuring light-evoked $Ca^{2+}$ responses in cone synaptic terminals (*Wei et al., 2012*). Animals were housed under a standard 12 hr day-night rhythm. Before the recordings, the mouse was dark-adapted for $\geq 1$ hr, then anaesthetized with isoflurane (Baxter, Unterschleißheim, Germany) and killed by cervical dislocation. The eyes were removed and hemisected in carboxygenated (95% $O_2$, 5% $CO_2$) artificial cerebrospinal fluid (ACSF) solution containing (in mM): 125 NaCl, 2.5 KCl, 2 $CaCl_2$, 1 $MgCl_2$, 1.25 $NaH_2PO_4$, 26 $NaHCO_3$, 20 glucose, and 0.5 L-glutamine (pH 7.4). The retina was separated from the eye-cup, cut in half, flattened, and mounted photoreceptor side-up on a nitrocellulose membrane (0.8 mm pore size, Merck Millipore, Darmstadt, Germany). Using a custom-made slicer (*Wei et al., 2012*; *Werblin, 1978*), acute vertical slices (200 μm thick) were cut parallel to the naso-temporal axis. Slices attached to filter paper were transferred on individual glass coverslips, fixed using high-vacuum grease and kept in a storing chamber at room temperature for later use. For imaging, individual retinal slices were transferred to the recording chamber of the 2P microscope (see below), where they were continuously perfused with warmed (36°C), carboxygenated extracellular solution.

For the **zebrafish larvae** experiments, we used 7 day post fertilisation (*dpf*) larvae of the zebrafish (*Danio rerio*) line *tg(1.8ctbp2:SyGCaMP6f)*, which expresses the genetically encoded $Ca^{2+}$ indicator GCaMP6f fused with synaptophysin under the RibeyeA promoter and allows measuring light-evoked $Ca^{2+}$ responses in bipolar cell synaptic terminals (*Dreosti et al., 2009*; *Johnston et al., 2019*; *Rosa et al., 2016*; *Zimmermann et al., 2018*). Animals were grown from 10 hr post fertilisation (*hpf*) in 200 μM of 1-phenyl-2-thiourea (Sigma) to prevent melanogenesis (*Karlsson et al., 2001*). Animals were housed under a standard 14/10 hr day-night rhythm and fed 3x a day. Before the recordings, zebrafish larvae were immobilised in 2% low-melting-point agarose (Fischer Scientific, Loughborough, UK; Cat: BP1360-100), placed on a glass coverslip and submersed in fish water. To prevent eye movement during recordings, α-bungarotoxin (1 nl of 2 mg/ml; Tocris, Bristol, UK; Cat: 2133) was injected into the ocular muscles behind the eye.

## Two-photon imaging

For all imaging experiments, we used MOM-type two-photon (2P) microscopes (designed by W. Denk, MPI, Heidelberg; purchased from Sutter Instruments/Science Products, Hofheim, Germany). For image acquisition, we used custom software (ScanM by M. Müller, MPI Neurobiology, Munich, and T.E.) running under IGOR Pro 6.3 for Windows (Wavemetrics, Lake Oswego, OR). The microscopes were equipped each with a mode-locked Ti:Sapphire laser (MaiTai-HP DeepSee, Newport Spectra-Physics, Darmstadt, Germany; or Chameleon Vision-S, Coherent; Ely, UK), two fluorescence detection channels for eCFP (FRET donor; HQ 483/32, AHF, Tübingen, Germany) and citrine (FRET acceptor; HQ 538/50, AHF) or GCaMP6f (ET 525/70 or ET 525/50, AHF), and a water immersion objective (W Plan-Apochromat 20×/1.0 DIC M27, Zeiss, Oberkochen, Germany). The excitation laser was tuned to 860 nm and 927 nm for TN-XL (eCFP) in mouse and GCaMP6f in zebrafish, respectively. Time-lapsed image series were recorded with 64 × 16 pixels (at 31.25 Hz) or 128 × 64 (at 15.625 Hz). Detailed descriptions of the setups for mouse (*Euler et al., 2019*; *Euler et al., 2009*; *Franke et al., 2017*) and zebrafish (*Zimmermann et al., 2018*) have been published elsewhere.

## Data analysis

Data analysis was performed using IGOR Pro (Wavemetrics). Regions of interest (ROIs) of individual synaptic terminals (of mouse cones and zebrafish bipolar cells) were manually placed. Then, $Ca^{2+}$ traces for each ROI were extracted for mouse cones as $\Delta R/R$, with the ratio $R = F_A/F_D$ of the FRET acceptor (citrine) and donor (eCFP) fluorescence, and resampled at 500 Hz. For zebrafish bipolar

cells, $Ca^{2+}$ traces for each ROI were extracted and detrended by high-pass filtering above ~0.1 Hz, followed by z-normalisation based on the time interval 1–6 s at the beginning of recordings using custom-written routines under IGOR Pro. A stimulus synchronisation marker that was generated by the visual stimulation software (Results) and embedded in the recordings served to align the $Ca^{2+}$ traces relative to the stimulus with ≤2 ms precision (depending on the scan line duration, see Results and *Euler et al., 2019*). For this, the timing for each ROI was corrected for sub-frame time-offsets related to the scanning.

*Response quality index.* To measure how well a cell responded to the sine wave stimulus, we computed the signal-to-noise ratio

$$Qi = \frac{Var\big[(C)_r\big]_t}{\big(Var[C]_t\big)_r} \tag{11}$$

where $C$ is the $T$ by $R$ response matrix (time samples by stimulus repetitions), while $()_x$ and $Var[]_x$ denote the mean and variance across the indicated dimension, respectively (*Baden et al., 2016*; *Franke et al., 2017*). For further analysis, we used only cells that had a $Qi>0.3$.

*Spectral contrast.* The mean trace in response to the green and UV sine wave stimulus was used to analyse the spectral sensitivity of the cones. For that, we computed the power spectrum of the trace and used the power ($P$) at the fundamental frequency (1 Hz) as a measure of response strength. Then, the spectral contrast ($SC$) was estimated as

$$SC = \frac{P_G - P_B}{P_G + P_B}, \tag{12}$$

where $P_G$ and $P_B$ correspond to the responses to green and UV, respectively. For statistical comparison of $SC$ values with and without silent substitution (see above), we used the Wilcoxon signed-rank test for non-parametric, paired samples.

## Acknowledgements

This study is part of the research program of the Bernstein Centre for Computational Neuroscience, Tübingen, and was funded by the German Federal Ministry of Education and Research and the Max Planck Society (BMBF, FKZ: 01GQ1002, and MPG M.FE.A.KYBE0004 to KF), the European Research Council (ERC-StG 'NeuroVisEco' 677687 to TB) the EU's Horizon 2020 research and innovation programme under the Marie Skłodowska-Curie grant agreement No 674901 ('switchBoard', to TB, TE), the UKRI (BBSRC, BB/R014817/1 and MRC, MC_PC_15071 to TB), the Leverhulme Trust (PLP-2017–005 to TB), the Lister Institute for Preventive Medicine (to TB), and the Deutsche Forschungsgemeinschaft (DFG, German Research Foundation; Projektnummer 276693517 – SFB 1233 to TE).

## Additional information

### Funding

| Funder | Grant reference number | Author |
|---|---|---|
| Bundesministerium für Bildung und Forschung | FKZ: 01GQ1002 | Katrin Franke |
| Max-Planck-Gesellschaft | M.FE.A.KYBE0004 | Katrin Franke |
| European Commission | ERC-StG 'NeuroVisEco' 677687 | Tom Baden |
| Horizon 2020 Framework Programme | Marie Skłodowska-Curie grant agreement No 674901 | Tom Baden Thomas Euler |
| Biotechnology and Biological Sciences Research Council | BB/R014817/1 | Tom Baden |
| Leverhulme Trust | PLP-2017-005 | Tom Baden |
| Lister Institute of Preventive Medicine | | Tom Baden |

| Deutsche Forschungsge-meinschaft | Projektnummer 276693517 - SFB 1233 | Thomas Euler |
|---|---|---|
| Medical Research Council | MC_PC_15071 | Tom Baden |

The funders had no role in study design, data collection and interpretation, or the decision to submit the work for publication.

### Author contributions

Katrin Franke, Conceptualization, Supervision, Funding acquisition, Investigation, Visualization, Methodology, Writing—original draft, Writing—review and editing; André Maia Chagas, Conceptualization, Validation, Methodology, Writing—review and editing; Zhijian Zhao, Conceptualization, Validation, Visualization, Methodology, Writing—review and editing; Maxime JY Zimmermann, Conceptualization, Validation, Investigation, Visualization, Writing—review and editing; Philipp Bartel, Yongrong Qiu, Klaudia P Szatko, Validation, Methodology; Tom Baden, Conceptualization, Funding acquisition, Visualization, Writing—review and editing; Thomas Euler, Conceptualization, Software, Supervision, Funding acquisition, Visualization, Methodology, Writing—original draft, Writing—review and editing

### Author ORCIDs

André Maia Chagas ⓘ https://orcid.org/0000-0003-2609-3017
Zhijian Zhao ⓘ http://orcid.org/0000-0002-3302-1495
Maxime JY Zimmermann ⓘ https://orcid.org/0000-0002-0885-9640
Philipp Bartel ⓘ https://orcid.org/0000-0002-7191-9511
Tom Baden ⓘ https://orcid.org/0000-0003-2808-4210
Thomas Euler ⓘ https://orcid.org/0000-0002-4567-6966

### Ethics

Animal experimentation: All animal procedures adhered to the laws governing animal experimentation issued by the German Government (mouse) or all procedures were performed in accordance with the UK Animals (Scientific Procedures) act 1986 and approved by the animal welfare committee of the University of Sussex (zebrafish larvae).

### Decision letter and Author response

Decision letter https://doi.org/10.7554/eLife.48779.023
Author response https://doi.org/10.7554/eLife.48779.024

## Additional files

### Supplementary files

• Supplementary file 1. Parts list of the through-the-objective mouse stimulator (*cf. Figure 2—figure supplement 1*; entries in grey are not shown).
DOI: https://doi.org/10.7554/eLife.48779.019
• Transparent reporting form
DOI: https://doi.org/10.7554/eLife.48779.020

### Data availability

Part lists are provided in Tables 1-3 and Supplementary file 1. Software scripts for stimulus calibration as well as design files for circuit boards and 3D-printed parts are provided at https://github.com/eulerlab/open-visual-stimulator (copy archived at https://github.com/elifesciences-publications/open-visual-stimulator). The visual stimulation software is provided at https://github.com/eulerlab/QDSpy (copy archived at https://github.com/elifesciences-publications/QDSpy).

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
