## [Decision Letter]

Thank you for submitting your article "An arbitrary-spectrum spatial visual stimulator for vision research" for consideration by *eLife*. Your article has been reviewed by three peer reviewers, one of whom is a member of our Board of Reviewing Editors, and the evaluation has been overseen by Joshua Gold as the Senior Editor. The following individuals involved in review of your submission have agreed to reveal their identity: Armin Bahl (Reviewer #2).

The reviewers have discussed the reviews with one another and the Reviewing Editor has drafted this decision to help you prepare a revised submission.

General:

The manuscript "An arbitrary-spectrum spatial visual stimulator for vision research" by Franke and colleagues describes the design of a new type of visual stimulator for vision neuroscience. The basic idea of this study is to use the increasingly popular, commercially available LightCrafter system in combination with a set of precisely tuned and filtered LEDs. Such an arrangement provides optimal experimental control over the color spectrum and temporal properties of the visual stimulus, features that are fundamentally needed for studies of color vision and generally helpful when visual stimulation is combined with fluorescent microscopy. The most attractive feature of this system is that it can be adapted to any spectral channels desired by the experimenter, and it can be tightly controlled to avoid visual stimulation during fluorescence image acquisition.

The authors provide a rigorous and detailed description of their system, including measurements of brightness gamma correction, spatial and temporal resolution, and chromatic aberration. Further, the authors test their visual stimulator under three different configurations and in two example biological applications, one in mouse retina explants and one in in vivo zebrafish larvae, showing that the visual stimulator behaves as expected. While modified LightCrafters have been extensively used in multiple previous studies in different model organisms, a detailed assembly manual for the use in visual research has been missing so far. In particular, the description of a two-projector system, allowing for up 6 arbitrary color channels to be presented, is novel and exciting. The work, therefore, is of general interest to the visual neuroscience community as a whole and should provide significant help for other researchers planning to implement such systems and adapt them to their specific needs. This constitutes a great service to the field.

Specific:

1) Because their system relies on commercially available LightCrafters, the authors should introduce them earlier in their manuscript, already in the Introduction (for example after the fourth paragraph). Here, the authors should also acknowledge and cite other studies (mouse retina, flies, zebrafish,.…) that have modified such systems by replacing LEDs and adding filters to improve spectral specificity. Moreover, when introducing this system, the authors should also mention that UV-optimized LightCrafters exist commercially. Moreover, it is important to note that the LightCrafter sequentially activates the different color LEDs (this is already described later in the manuscript, but it would help to have this in the Introduction. For example, the authors could move the opening sentence of the fourth paragraph of subsection “Separating light stimulation and fluorescence detection” into the Introduction). Finally, it would be nice to know if other studies in systems neuroscience have already used the fiber-coupled version of the LightCrafter.

2) In the description of the stimulator, it does not become clear whether it is at all possible to project stimuli that consist of spatially patterned multi-chromatic light, for example a blue-green square-wave grating. Since the electronics of the LCr control the timing of the LEDs, this arrangement should allow for the described external light engine (when controlled by the LCr electronics) to provide multispectral stimulation. It would be important to point out explicitly how this can be realized with the described system.

3) For readers wishing to rebuild such a system, I suspect that the most challenging part is the printing and soldering of the circuit boards controlling the LEDs. Here, more information about these boards would help. The authors seem to have uploaded their designs as Gerber files to the provided github folder. Could the authors explain (briefly in the text and in more detail in the github readme.md) how to load these files and how to send them to manufacturing? What software did the authors use for the design, what software (open-source) do they recommend to the reader? Additionally, it would be nice to have an actual image of the PCB board drawings in a supplementary figure as well as a table of the needed electrical components.

4) The authors say that the four-channel zebrafish stimulator provides a more general solution. When does the simpler stimulator fail? Would the latter circuit board design also be fine with the two-channel mouse stimulator? Is it correct to conclude that for the four-channel zebrafish stimulator, one would need 2 boards as in Figure 2—figure supplement 5F and 4 boards as in 5G? What external power supply and what voltage do the authors drive their LEDs with in the two circuit board designs, or are all LEDs always powered by the LightCrafter?

5) Please provide a more detailed drawing and explanation of the collimator system and add the missing parts to Table 2. What holds the LEDs firmly in place and are they cooled via fans? What holds the bandpass filters in place, etc.? Are some of the parts 3d-printed? If so, which ones?

6) The authors have characterized the spatial properties of the "through-objective-configuration" but have not done this for the "through-the-collimator" configuration and for the side projection in the zebrafish preparation. I assume they have focused only on the former configuration as passing light patterns through the objective might lead to unexpected distortions while in the latter two configurations, this is less of a problem. Is this true? If so, the authors should say why such quantification has only been done for the through-the-objective configuration. Otherwise, please provide respective measurements.

7) Is the ScanM software for the microscope from Sutter or is it self-written by the group? According to their website, the Sutter MOM microscopes come with a software called MScan. How are these two software packages related, or is this the same? Please provide a link to this software if it is open-source or point to the Sutter website.

8) Do the authors gate the PMTs during the retrace period in order to protect the PMTs? Would this actually be necessary? If yes, is this automatically done by the MOM microscope software? The authors further make a point that scan patterns like spiral scanning or back scanning patterns have little or no retrace period (subsection “Separating light stimulation and fluorescence detection”). The authors should elaborate on how the temporal separation would work without retrace periods?

9) The authors talk about application of their visual stimulator for studying retinal explants and larval zebrafish vision. Could the authors also discuss the use of their system in other models, such as *Drosophila*? Perhaps the authors could provide *Drosophila* photoreceptor spectra in the supplement and describe what changes would be needed in the zebrafish system to optimize the design for *Drosophila*? If feasible, the authors might even provide a more generalized version of the zebrafish calibration ipython notebook?

10) The authors nicely show the concept of the silent substitution protocol in the mouse preparation. Because of the 4 cone types in zebrafish, whose spectra are significantly more overlapping, such experiments are likely more complicated. In Figure 6, how do you make sure that UV is not just activating all cone types? Could the authors discuss that a silent substitution protocol as was done for the mouse retina could increase confidence about their UV specificity?

11) For the zebrafish stimulator system, one major additional advantage is that one can quickly modify the visual stimulus spectrum to separate it maximally from fluorescent probe detection. If one would like to image green probes, one could not use the green LED but all other LEDs. If one would like to image red probes, one could turn off the red LED and leave all other LEDs on. This makes the system more flexible compared to what is commonly used in the field (fixed wavelength, often red, and difficult to change within an experimental session) and interesting for researchers wanting to occasionally image red-shifted calcium indictors in the same microscope where they normally use GCaMP. Furthermore, their system allows one to fully turn off stimulation through software to study the circuit in complete darkness. In contrast, commonly used projector systems show always some residual light even when one sends black to the projector or monitor. Discussing these points would make their system even more interesting to a broad readership.

12) The authors mention that QDSpy generates a "compiled" version of the stimulus (subsection “Visual stimulation software”). What does this mean? Do the authors upload a "movie pattern" to the LightCrafter, which is then displayed in a loop?

13) In Figure 3F, are both peaks and colors s/cones? Where is the m/cone?

14) In Figure 2—figure supplement 2, the b-label should be in the second line of panels. The authors say that the "ringing" comes aliasing-related fluctuations with the 60 Hz projector (subsection “Separating light stimulation and fluorescence detection”) but could it also come from the on/off dynamics of the LED switching? Which circuit board design was used here, the one in Figure 2—figure supplement 5C or the one Figure 2—figure supplement 5F,G? In the latter, would the dynamics look similar? If the authors have measured this already, it would be nice to see this in the supplement, or at least mention it.

15) The authors discuss the possibility to use mechanical choppers as blanking signals (subsection “Potential issues and technical improvements”). Why is the design of the "blanking" circuits more demanding when LED power is higher? Because of on/off dynamics of the LEDs? Are there systems neuroscience papers that the authors can cite that have already used mechanical choppers during fluorescent imaging?

16) Subsection “Visual stimuli for current animal models”: This section on the range of photopigments in different species is interesting, but such detail is not really required in this context. If the authors wished to shorten the Introduction, this is a section that could be abridged a bit.

17) Subsection “Stimulator design” paragraph one: "beamer" is rather colloquial. Perhaps "digital projector"?

18) Subsection “Stimulator design”: ("small footprint"): The actual dimensions (i.e., L x W cm) would be a nice addition to give the reader an immediate idea of the size without poring through the TI literature.

19) Subsection “Stimulator design” paragraph two: ("coupled by a light guide"): It would be nice to have a bit more detail here about how the external LED input is introduced to the LCr. Presumably some internal optics need to be removed from the LCr, and an entry port must be fashioned. How critical is the alignment of the incoming signal?

20) It is not clear, why spectral separation of visual stimulation and fluorescence detection is necessary, if the temporal separation approach works. In general, it seems that spectral separation reduces the claimed versatility of the stimulator. Related to that, Figure 2—figure supplement 3 is very hard to understand. The curves should be labeled better. Second, it seems like for the zebrafish-stimulator the green LED is not transmitted at all by the dichroic mirror.

21) The discussion of the paper should provide an outlook on potential applications that require the design of this stimulator.

---

## [Author Response]

Specific1) Because their system relies on commercially available LightCrafters, the authors should introduce them earlier in their manuscript, already in the Introduction (for example after the fourth paragraph). Here, the authors should also acknowledge and cite other studies (mouse retina, flies, zebrafish,.…) that have modified such systems by replacing LEDs and adding filters to improve spectral specificity. Moreover, when introducing this system, the authors should also mention that UV-optimized LightCrafters exist commercially. Moreover, it is important to note that the LightCrafter sequentially activates the different color LEDs (this is already described later in the manuscript, but it would help to have this in the Introduction. For example, the authors could move the opening sentence of the fourth paragraph of subsection “Separating light stimulation and fluorescence detection” into the Introduction). Finally, it would be nice to know if other studies in systems neuroscience have already used the fiber-coupled version of the LightCrafter.

Good point. We followed the reviewers’ suggestion and introduce the working principle of a DMD-based LCr already in the Introduction of the revised manuscript. Also, we now acknowledge earlier studies that employed (modified) DLP projectors in different species.

As to the question if other studies have used fibre-coupled systems — as far as we know, there are only a few studies that use a light guide (e.g. Tan et al., 2015, which we now cite in the revised manuscript). It is possible though, that this information is not always given in the Materials and methods of such publications.

Finally, we added a new table (Table 4) that lists some of the commercially available DLP projectors with custom LEDs.

2) In the description of the stimulator, it does not become clear whether it is at all possible to project stimuli that consist of spatially patterned multi-chromatic light, for example a blue-green square-wave grating. Since the electronics of the LCr control the timing of the LEDs, this arrangement should allow for the described external light engine (when controlled by the LCr electronics) to provide multispectral stimulation. It would be important to point out explicitly how this can be realized with the described system.

We are not sure if we understand the reviewers’ question correctly.

In the experiments described in the manuscript, we run the LCr(s) in “video mode”, which means that we feed a normal PC output to the LCr via its HDMI connector. Here, an LCr behaves like a monitor. In this mode, each color channel in an R-G-B image (3x8 bits) can be assigned to one of the 3 LEDs via the QDSpy software — or, in the case of the dual-LCr zebrafish version, each color channel in an R-G-B-UV image to one of the 6 LEDs. Hence, for a blue-green square wave grating, one just needs to generate a B+G stimulus to enable both the green and the blue LED.

In “video mode”, it is not possible to assign a combination of LEDs to a single color channel. However, in the so-called “pattern mode”, each bitplane in a 3x8 R-G-B image can be assigned to an arbitrary combination of LEDs. Using the pattern mode is just a question of software; the hardware we describe in the manuscript (i.e. LED drivers, blanking etc) supports it. We recently updated the QDSpy stimulus software to give the user access to the pattern mode (http://qdspy.eulerlab.de/lightcrafter.html#example-scripts).

In the revised manuscript, we clarify the points above and refer to the pattern mode option in QDSpy in the Discussion and Table 1.

3) For readers wishing to rebuild such a system, I suspect that the most challenging part is the printing and soldering of the circuit boards controlling the LEDs. Here, more information about these boards would help. The authors seem to have uploaded their designs as Gerber files to the provided github folder. Could the authors explain (briefly in the text and in more detail in the github readme.md) how to load these files and how to send them to manufacturing? What software did the authors use for the design, what software (open-source) do they recommend to the reader? Additionally, it would be nice to have an actual image of the PCB board drawings in a supplementary figure as well as a table of the needed electrical components.

In addition to clarifying which board(s) are needed for which stimulator version (see the reply to question #4), we added more details on the boards to the Github repository [https://github.com/eulerlab/open-visual-stimulator], including information on the software used to design the boards (KiCad 5, now also in the Key Resource Table = Table 1), how to order these boards (link to a video), a selection of companies that offer on-demand printed circuit board service, and the electronic parts needed. The latter are provided (in the repository) for each board as a bill-of-material (BOM) file in the “cvs” (comma-separated value) format; this is a common industry standard. Since sometimes, electronic parts need to be replaced when they are no longer produced or because better, pin-compatible parts become available, we prefer to keep these BOMs in the repository (instead of adding them as tables to the manuscript).

As requested by the reviewers, we added (rendered) images of the boards to Figure 2—figure supplement 5.

4) The authors say that the four-channel zebrafish stimulator provides a more general solution. When does the simpler stimulator fail? Would the latter circuit board design also be fine with the two-channel mouse stimulator? Is it correct to conclude that for the four-channel zebrafish stimulator, one would need 2 boards as in Figure 2—figure supplement 5F and 4 boards as in 5G? What external power supply and what voltage do the authors drive their LEDs with in the two circuit board designs, or are all LEDs always powered by the LightCrafter?

The simple solution (based on the board in Figure 2—figure supplement 5A-C) would “fail” when the LED currents required are higher than 0.75 A, which is what the three parallel optocouplers on the board can pass (3x 250 mA continuous current load). Here, the LCr’s built-in drivers are not limiting, as they can provide up to 4.3 A in total. The more general solution (based on a combination of boards; see Figure 2—figure supplement 5D-H) used in the zebrafish stimulator is also compatible with the mouse stimulator. The main advantage of this second solution is that it does not rely on the limited current from the LCr’s built-in LED drivers to run the external LED. Instead, it can use any external current supply and, therefore, is very flexible in terms of LED choice (including high-power LEDs) - in this sense, it is more general. For LED timing, the more general solution makes use of the same digital signals that the LCr uses for its built-in drivers; these signals are available on the LCr’s motherboard (Figure 2—figure supplement 5H).

The board (Figure 2—figure supplement 5A-E) the simple solution relies on supports up to 4 LED channels (in the figure, only the components for two channels are soldered). Therefore, both for the mouse and the zebrafish stimulator one needs just a single board.

The more general solution relies on a board that combines LCr and microscope signals (logic board w/ up to 3 LED channels; Figure 2—figure supplement 5F) and on a current driver board for the LED (only 1 LED channel; Figure 2—figure supplement 5G). Therefore, for the mouse stimulator, one would need 1 logic board and 2 driver boards. For the zebrafish stimulator, one needs 2 logic boards and 4 driver boards.

The legend of Figure 2—figure supplement 5 and the text was updated to reflect the points discussed above.

5) Please provide a more detailed drawing and explanation of the collimator system and add the missing parts to Table 2. What holds the LEDs firmly in place and are they cooled via fans? What holds the bandpass filters in place, etc.? Are some of the parts 3d-printed? If so, which ones?

We now show more details about the individual parts of the collimator system in Figure 2—figure supplement 6 and added the parts to Table 2. Also, we added the files for the custom built parts to the GitHub repository. In addition, we clarify in the legends of the respective figures that we do not cool the LEDs with fans.

6) The authors have characterized the spatial properties of the "through-objective-configuration" but have not done this for the "through-the-collimator" configuration and for the side projection in the zebrafish preparation. I assume they have focused only on the former configuration as passing light patterns through the objective might lead to unexpected distortions while in the latter two configurations, this is less of a problem. Is this true? If so, the authors should say why such quantification has only been done for the through-the-objective configuration. Otherwise, please provide respective measurements.

We had focussed on quantifying the spatial resolution of the mouse stimulator using the “through-the-objective” (TTO) configuration more for practical reasons, as this allowed to magnify the image projected on the Raspberry Pi camera chip using a 5x objective — instead of the (higher quality) 20x objective normally used for functional recordings. This ensured that the spatial resolution of the camera itself was not the limiting factor.

Pictures taken during routine stimulator alignment suggested that the “through-the-condenser” (TTC) configuration has a similar performance. For the revised manuscript, we quantified the spatial resolution of the TTC configuration as well, again by projecting UV and green checkerboards onto a Raspberry Pi camera chip. We found that both contrast and sharpness of transitions between bright and dark checkers declined for somewhat larger checker sizes compared to the TTO configuration (5-10 μm vs. 4 μm). This slightly lower performance of the TTC configuration is likely related to the fact that the image projected onto the camera chip was not magnified (see above) and therefore reached the resolution limit of the camera.

We have added the new measurements to Figure 4 and mention the findings in the respective Results section.

For the zebrafish stimulator, a similar approach for quantifying its spatial resolution is possible (Author response image 1). However, as checker sizes at the limit of the zebrafish ´s spatial resolution (2° visual angle (Haug et al., 2010)) are relatively large (~1 mm on the teflon screen for our setup), the spatial resolution of the LCr is not a limiting factor.

**Author response image 1. respfig1:** Images of checkerboard stimuli with varying sizes for UV LED of the zebrafish stimulator, recorded using a Raspberry Pi camera positioned between the LCrs and the teflon screen.

7) Is the ScanM software for the microscope from Sutter or is it self-written by the group? According to their website, the Sutter MOM microscopes come with a software called MScan. How are these two software packages related, or is this the same? Please provide a link to this software if it is open-source or point to the Sutter website.

ScanM is a scanning software for Sutter Instruments’ MOM. The development started in the Dep. Of Winfried Denk at the MPI for Medical Research in Heidelberg, Germany, and was further developed by M. Müller (MPI Neurobiology, Munich) and T.E. in Tübingen. While the source code is not proprietary, we did not publish it yet because it is so far only used by the labs of K.F., T.B and T.E., it runs on IGOR Pro, and the authors cannot provide larger scale support. In any case, the microscope software is not relevant for the stimulator solutions presented here. The blanking signal used for switching off the LEDs during data acquisition can also be generated by other 2P microscope

software packages.

8) Do the authors gate the PMTs during the retrace period in order to protect the PMTs? Would this actually be necessary? If yes, is this automatically done by the MOM microscope software? The authors further make a point that scan patterns like spiral scanning or back scanning patterns have little or no retrace period (subsection “Separating light stimulation and fluorescence detection”). The authors should elaborate on how the temporal separation would work without retrace periods?

We did not use gated PMTs. While these may further reduce the stimulus artifact and possibly prolong the PMTs’ lifetime, we did not consider gated PMTs absolutely necessary, because with our approach, light artifacts are usually manageable (they only appear in 1-2 pixel columns in the recorded images and are cut off) and so far PMTs did not die on us too often. In fact, we did test at some point if we can gate our standard GaAsP PMTs quickly enough (by modulating the voltage that determines the gain) but were unsuccessful.

The temporal separation concept we describe relies on frequent LED-on periods, e.g. 20% of a scan line (like 400 μs per 2 ms line). In x-y scans, we use the “retrace” as LED-on period. If other scan types are used, one would have to design them such that LEDs can be frequently turned on (during times when no data are collected).

As to the retrace (or the potential lack thereof) in spiral scanning: Maybe the term “spiral” is confusing here. There are different ways of designing smooth scan trajectories that reduce the need for sharp turns or avoid these altogether. Therefore, these may not need retrace periods. We recently published a paper (Rogerson et al., 2019), where we presented a scan configuration consisting of a sequence of angular-offset, curved trajectories that form a spiral-like pattern. This pattern, which we dubbed “spiral scan”, allows scanning at twice the temporal and twice the spatial resolution compared to our “standard” x-y scans. In these scans, we added a section per line (dubbed “retrace”) for switching the LEDs on.

These points are now reflected/clarified in the revised manuscript. However, we think that details on the design of suitable “spiral scans” are beyond the scope of the manuscript, in particular as they are discussed elsewhere (e.g. Rogerson et al., 2019; Euler et al., 2019).

9) The authors talk about application of their visual stimulator for studying retinal explants and larval zebrafish vision. Could the authors also discuss the use of their system in other models, such as Drosophila? Perhaps the authors could provide Drosophila photoreceptor spectra in the supplement and describe what changes would be needed in the zebrafish system to optimize the design for Drosophila? If feasible, the authors might even provide a more generalized version of the zebrafish calibration ipython notebook?

We thank the reviewers for these suggestions. To illustrate the use of our system for other model species like *Drosophila*, we have made the following changes to the manuscript:

1) We have modified the zebrafish calibration notebook to match the more general design of the mouse calibration notebook. By doing so, both calibration files can now be easily adapted to the spectral sensitivities of other species. We also mention the flexibility of these notebooks in the respective Results section.

2) Throughout the manuscript, we now indicate if our solutions are specific to mouse and/or zebrafish, or may also be suitable for other model species.

3) We added Figure 2—figure supplement 4 with possible LED/filter combinations matching the spectral sensitivity of *Drosophila*. These can be used in combination with the dichroic mirror designed for the zebrafish stimulator.

10) The authors nicely show the concept of the silent substitution protocol in the mouse preparation. Because of the 4 cone types in zebrafish, whose spectra are significantly more overlapping, such experiments are likely more complicated. In Figure 6, how do you make sure that UV is not just activating all cone types? Could the authors discuss that a silent substitution protocol as was done for the mouse retina could increase confidence about their UV specificity?

Due to the “sensitivity tail” of all opsins for shorter wavelengths (“β-band”), the UV LED/filter combination used in the zebrafish stimulator is indeed activating all cone types (for numbers, see calibration file on GitHub). However, the UV LED is still activating UV-opsin stronger than all other opsin types (e.g. 82% and 65% for UV- and S-opsin). Therefore, by expressing chromatic preference as a function of relative activation by the four LED channels, this imperfect spectral separation is sufficient to differentiate between e.g. a UV- and blue-preferring cell.

We now mention in the respective Results section that spectral separation can be improved by using a silent substitution protocol. In addition, we added a notebook to the GitHub repository illustrating how silent substitution for zebrafish could be implemented.

11) For the zebrafish stimulator system, one major additional advantage is that one can quickly modify the visual stimulus spectrum to separate it maximally from fluorescent probe detection. If one would like to image green probes, one could not use the green LED but all other LEDs. If one would like to image red probes, one could turn off the red LED and leave all other LEDs on. This makes the system more flexible compared to what is commonly used in the field (fixed wavelength, often red, and difficult to change within an experimental session) and interesting for researchers wanting to occasionally image red-shifted calcium indictors in the same microscope where they normally use GCaMP. Furthermore, their system allows one to fully turn off stimulation through software to study the circuit in complete darkness. In contrast, commonly used projector systems show always some residual light even when one sends black to the projector or monitor. Discussing these points would make their system even more interesting to a broad readership.

We thank the reviewers for these suggestions. In the revised manuscript, we now mention these points in the Results section “LED selection and spectral calibration”.

12) The authors mention that QDSpy generates a "compiled" version of the stimulus (subsection “Visual stimulation software”). What does this mean? Do the authors upload a "movie pattern" to the LightCrafter, which is then displayed in a loop?

We agree with the reviewers that this sentence is unclear. To answer the reviewers’ question: QDSpy does not generate a sequence of images that is uploaded to the flash memory (EEPROM) of the LCr board. QDSpy drives the LCr as if it was a normal 60 Hz display (in “video mode”, see also our reply to question #2).

QDSpy stimuli are written as normal Python scripts that use a specific Python library to generate different stimulus elements (e.g. flashing spots, moving gratings or bars, movies). In these scripts, everything is allowed, including complex calculations. Therefore, without restricting the user in how to write the stimulus code, it cannot be guaranteed that the script runs fast enough to generate stimulus frames at 60 Hz (that is, without “dropping” frames). To ensure stimulus timing, the first time QDSpy runs a stimulus script, it generates a boiled-down (“compiled”) version of the stimulus that is stored in a separate file. When the user runs that stimulus again (and the source Python script of the stimulus has not been altered after compilation), QDSpy presents the stimulus from the compiled file. This compiled file contains the drawing instructions and timing for every stimulus element used in a very compact form, not unlike OpenGL graphic commands. The advantage is, that any more time consuming calculation has already been carried out at compile time, hence stimulus timing is typically guaranteed (at least most of the time, because it is still Windows, with tons of background activity, and not a real-time operating system). The main disadvantage of this strategy is that the possibilities of influencing the stimulus while it runs are (currently) very limited.

We now clarify the term “compiled” in the respective section of the revised manuscript.

13) In Figure 3F, are both peaks and colors s/cones? Where is the m/cone?

Thanks for pointing this out. We define isomerisation rate as photoisomerisations (P*) per second (s) and cone. Therefore, the “s/cones” does not indicate “S-cone”, but instead “seconds/cone”. We now explain this abbreviation in the legend of Figure 3F.

14) In Figure 2—figure supplement 2, the b-label should be in the second line of panels. The authors say that the "ringing" comes aliasing-related fluctuations with the 60 Hz projector (subsection “Separating light stimulation and fluorescence detection”) but could it also come from the on/off dynamics of the LED switching? Which circuit board design was used here, the one in Figure 5—figure supplement 2C or the one Figure 5—figure supplement 2F,G? In the latter, would the dynamics look similar? If the authors have measured this already, it would be nice to see this in the supplement, or at least mention it.

Thanks; we revised Figure 2—figure supplement 2 as detailed below.

As for the photodiode recordings: We used the circuit board design shown in Figure 2—figure supplement 5C. As mentioned in the Results section, the slow intensity fluctuations at ~1 Hz apparent in the chirp trace (revised Figure 2—figure supplement 2B) likely reflect aliasing, because the LCr frame rate (60 Hz) and the blanking/LED-on signal (typically 500 Hz) were not synchronized.

The fast “ringing” in the traces shown in Figure 2—figure supplement 2A,B, on the other hand, is probably a recording artifact introduced by the circuit we devised to read out the photodiode. To confirm this, we used a PMT instead of the photodiode to look at the fine structure of the LED pulses and did not observe any “ringing”. Hence, we replaced former panels A,B by the new recordings.

15) The authors discuss the possibility to use mechanical choppers as blanking signals (subsection “Potential issues and technical improvements”). Why is the design of the "blanking" circuits more demanding when LED power is higher? Because of on/off dynamics of the LEDs? Are there systems neuroscience papers that the authors can cite that have already used mechanical choppers during fluorescent imaging?

In general, designing an electronic circuit that switches high loads (currents) at a fast timescale with short rise and decay times is more challenging than one with a lower load. For example, the simple board (Figure 2—figure supplement 5A-C; see our reply to question #4) contains only very few components (mainly fast solid-state relays) but can switch “only” up to 0.75 A. The LED driver board (Figure 2—figure supplement 5G; see our reply to question #4) is more complex and can switch up to 10 A. With LED technology booming and new integrated LED driver chips being developed, designing these boards likely become less of a challenge in the future. Nevertheless, we felt that the chopper solution should be mentioned for completeness sake.

The respective section of the revised manuscript has been clarified. In addition, we now cite work that employed choppers.

16) Subsection “Visual stimuli for current animal models”: This section on the range of photopigments in different species is interesting, but such detail is not really required in this context. If the authors wished to shorten the Introduction, this is a section that could be abridged a bit.

We thank the reviewers for this suggestion. We decided to keep this section as it nicely illustrates the importance of matching the visual stimulation to the highly variable spectral sensitivities of common model systems used in visual neuroscience.

17) Subsection “Stimulator design” paragraph one: "beamer" is rather colloquial. Perhaps "digital projector"?

Changed.

18) Subsection “Stimulator design”: ("small footprint"): The actual dimensions (i.e., L x W cm) would be a nice addition to give the reader an immediate idea of the size without poring through the TI literature.

We now added the dimensions of the LCr to the text.

19) Subsection “Stimulator design” paragraph two: ("coupled by a light guide"): It would be nice to have a bit more detail here about how the external LED input is introduced to the LCr. Presumably some internal optics need to be removed from the LCr, and an entry port must be fashioned. How critical is the alignment of the incoming signal?

The Fiber-E4500MKIITM (EKB) is fit by the company (EKB) with a built-in port for standard light guides (7 mm outer diameter, 5 mm core diameter; see recommendations by EKB); the LCr’s internal optics are also already modified accordingly. In our experience, no further alignment is required when connecting the light guide to the LCr port. We think it is possible to convert a standard LED-equipped LCr, but we have not yet tried that.

This information was now added to the manuscript.

20) It is not clear, why spectral separation of visual stimulation and fluorescence detection is necessary, if the temporal separation approach works. In general, it seems that spectral separation reduces the claimed versatility of the stimulator. Related to that, Figure 2—figure supplement 3 is very hard to understand. The curves should be labeled better. Second, it seems like for the zebrafish-stimulator the green LED is not transmitted at all by the dichroic mirror.

Even though LED stimulation and fluorescent data acquisition are temporally separated, spectral separation is needed to protect the PMTs from the stimulus light, because the PMTs are not switched off (gated) during the LED-on times (see our reply to question #8). While the stimulus light may not damage the PMTs (right away), it often triggers the overcurrent protection circuit many PMTs are equipped with, thereby shutting them off.

In a perfect world, the combination of filters in front of the LEDs and those in front of the PMTs would take care of any crosstalk (i.e. stimulus artefacts in the PMT signal), but because filters are never perfectly band-pass, stimulus light is relatively bright, and PMTs need to be extremely sensitive to pick up the fluorescence signal, crosstalk can, in our experience, be never completely avoided.

In addition to the combination of filters for LEDs and PMTs, DM M (or DM Z) — the custom dichroic mirror above the objective lens (cf. Figure 2—figure supplement 1) — plays a role in spectral separation, but this role depends on the stimulator configuration:

In TTO (through-the-objective) configuration, DM M (or DM Z) is crucial, because it allows the excitation laser and the stimulus light pass towards the specimen, while reflecting fluorescence from the specimen to the PMTs (and passing stimulus light reflected back from the specimen).

In TTC (through-the-condenser) configuration, the main role of DM M is to reflect fluorescence from the specimen to the PMTs and prevent any stimulus light reflected back from the specimen going there by passing it (similar to TTO).

In case of the zebrafish configuration with the teflon screen, some of the stimulus light is scattered in the probe towards the objective lens. Like for the TTC configuration, DM Z helps to reduce the amount of stimulus light entering the PMTs.

Taken together, in our experience (and for the presented applications), spectral separation is a necessity. However, we do not think it reduces the versatility of the stimulator design: First, there is a great variety of dichroic filters and LEDs commercially available, such that finding suitable and affordable combinations that enable spectral separation is not difficult. Second, as we illustrate in the new Figure 2—figure supplement 4, the already designed DMs could be used also for other species (e.g. DM Z for *Drosophila*).

With respect to Figure 2—figure supplement 3, we labeled the curves, as suggested by the reviewers. Further, we tried to improve the visibility of the curves.

It is true that DM Z and the green filter (Figure 2—figure supplement 3C) overlap only a little. However, with a high intensity LED, we still get sufficient stimulation light in this band. Note that this issue only occurs in case of the TTO configuration, where the stimulus light comes through the objective lense. For the zebrafish experiments shown in this study, the stimulus is projected at a Teflon screen, and therefore DM Z does not restrict the LED bands.

21) The discussion of the paper should provide an outlook on potential applications that require the design of this stimulator.

Thanks for this suggestion. We added a section to the Discussion (“Towards a common stimulator design for vision research”) where we discuss potential applications and highlight the advantages of our system compared to previous/commercially available ones.